# GP64-pseudotyped lentiviral vectors target liver endothelial cells and correct hemophilia A mice

Michela Milani [ID][1], Cesare Canepari[1,2], Simone Assanelli[3], Simone Merlin[3], Ester Borroni[3], Francesco Starinieri [ID][1], Mauro Biffi [ID][1], Fabio Russo[1], Anna Fabiano[1], Desirèe Zambroni [ID][4], Andrea Annoni [ID][1], Luigi Naldini [ID][1,2], Antonia Follenzi[3] & Alessio Cantore [ID][1,2]✉

## Abstract

**Lentiviral vectors (LV) are efficient vehicles for in vivo gene delivery to the liver. LV integration into the chromatin of target cells ensures their transmission upon proliferation, thus allowing potentially life-long gene therapy following a single administration, even to young individuals. The glycoprotein of the vesicular stomatitis virus (VSV.G) is widely used to pseudotype LV, as it confers broad tropism and high stability. The baculovirus-derived GP64 envelope protein has been proposed as an alternative for in vivo liver-directed gene therapy. Here, we perform a detailed comparison of VSV.G- and GP64-pseudotyped LV in vitro and in vivo. We report that VSV.G-LV transduced hepatocytes better than GP64-LV, however the latter showed improved transduction of liver sinusoidal endothelial cells (LSEC). Combining GP64-pseudotyping with the high surface content of the phagocytosis inhibitor CD47 further enhanced LSEC transduction. Coagulation factor VIII (FVIII), the gene mutated in hemophilia A, is naturally expressed by LSEC, thus we exploited GP64-LV to deliver a FVIII transgene under the control of the endogenous FVIII promoter and achieved therapeutic amounts of FVIII and correction of hemophilia A mice.**

**Keywords** In Vivo Gene Therapy; Lentiviral Vectors; Liver Endothelial Cells; Hemophilia A; Envelope Engineering
**Subject Categories** Genetics, Gene Therapy & Genetic Disease; Haematology

## Introduction

The liver is one of the most relevant target organs for in vivo gene therapy since it offers the possibility to treat several inherited metabolic disorders, including plasma protein deficiencies, such as hemophilia (Cantore et al, 2022; Zabaleta et al, 2019). Adeno-associated virus (AAV)-derived vectors and lentiviral vectors (LV) are both efficient vehicles for in vivo gene transfer to the liver, with the former being in an advanced phase of clinical development or

commercially available, while the latter has yet to reach clinical testing in the context of systemic administration (Cantore and Naldini, 2021). A single intravenous (i.v.) administration of AAV vectors delivering a functional clotting factor transgene showed prolonged therapeutic benefit in adult people with hemophilia (Mahlangu et al, 2023; Pipe et al, 2023). Meanwhile, systemic administration of lentiviral vectors (LV) showed stable liver gene transfer in animal models, such as mice, dogs, and non-human primates (NHP), and allowed phenotypic correction of hemophilia models (Cantore et al, 2015; Merlin et al, 2017; Milani et al, 2019, 2022; Nicolas et al, 2022). Compared to AAV vectors, LV integrate into the genome of transduced cells, potentially ensuring life-long gene transfer even after a single administration to pediatric patients and possess a larger packaging capacity. Moreover, the percentage of humans with anti-AAV neutralizing antibodies, which are generally excluded from receiving systemic AAV-vector gene therapy is higher than the prevalence of anti-LV immunity. For these reasons, LV may complement the reach of AAV-vector-mediated liver gene therapy in the future. Typically, the tropism of AAV vectors depends on the capsid, while that of LV is directed by the pseudotype envelope protein chosen. The surface glycoprotein of the vesicular stomatitis virus (VSV.G) is the most widely used envelope protein, as it confers broad tropism and high stability to LV particles. However, numerous alternative LV pseudotypes have been investigated (Frank and Buchholz, 2019; Gutierrez-Guerrero et al, 2020). We have previously shown that, upon i.v. administration, most VSV.G-pseudotyped LV (VSV.G-LV) distribute to the liver, where they transduce hepatocytes, liver macrophages (Kupffer cells, KC), and liver sinusoidal endothelial cells (LSEC) (Fama et al, 2021; Merlin et al, 2017, 2019; Milani et al, 2019). All these cell types are important targets for different therapeutic purposes, in particular, hepatocytes naturally produce coagulation factor IX (FIX), which is lacking in hemophilia B, while coagulation factor VIII (FVIII), whose lack causes hemophilia A, is physiologically produced by LSEC, among other sources (Follenzi et al, 2008; Zanolini et al, 2015). The baculovirus-derived GP64 envelope protein has been proposed as a possible alternative to VSV.G to pseudotype LV for use in in vivo liver-directed gene therapy (Schauber-Plewa et al, 2005; Schauber et al, 2004). Indeed, this envelope protein was previously reported to confer tropism restriction against hematopoietic-lineage cells in vitro to

[1]San Raffaele Telethon Institute for Gene Therapy, IRCCS San Raffaele Scientific Institute, Milan, Italy. [2]Vita-Salute San Raffaele University, Milan, Italy. [3]Department of Health Sciences, University of Piemonte Orientale, Novara, Italy. [4]IRCCS San Raffaele Scientific Institute, Milan, Italy. ✉E-mail: cantore.alessio@hsr.it

pseudotyped LV, and improved resistance to complement-mediated inactivation (Schauber-Plewa et al, 2005; Schauber et al, 2004). Both these aspects are important to achieve efficient in vivo gene transfer since vector particles are susceptible to complement-mediated inactivation and clearance by professional phagocytes after i.v. infusion (Milani et al, 2017, 2019). We have previously shown that LV with a high surface content of the natural phagocytosis inhibitor CD47 (CD47hi LV) are more resistant to capture by professional phagocytes resulting in improved gene transfer to hepatocytes (Milani et al, 2019). Thus, pseudotyping CD47hi LV with the GP64 envelope protein may further increase the transduction of target cells. Here, we perform an in-depth characterization of the potential advantages of using GP64-pseudotyped LV (GP64-LV) compared to VSV.G-LV for in vivo liver-directed gene therapy, by assessing the susceptibility to phagocytosis, the efficiency of gene transfer to different cell types, in vitro and in vivo. We then exploit GP64-LV to deliver a FVIII transgene to LSEC and achieve stable therapeutic levels of FVIII up to the normal range in hemophilia A mice, both treated as adults and as newborns.

# Results

## Evaluation of infectious titer and resistance to complement-mediated inactivation of GP64-LV

We first produced GP64-LV or VSV.G-LV with GFP or FIX as transgene and compared their infectious titers both in a reporter 293T cell line, commonly used to titer VSV.G-LV, and in Huh7, a human hepatocarcinoma cell line (Fig. 1A). In parallel, we measured physical vector particles, expressed as the concentration of HIV Gag p24, and calculated the specific infectivity, as the ratio of infectious to physical particles (Fig. 1B,C). Whereas infectious titers of VSV.G-LV did not differ between the two cell lines, we observed a 20-fold lower infectious titer of GP64-LV when determined on 293T compared to Huh7 cells (Fig. 1D), in line with a previous report (Schauber et al, 2004). Thus, we decided to use the titer measured on the Huh7 cell line, to match input doses of the two differently pseudotyped LV. Since VSV.G-LV similarly transduced both 293T and Huh7 cells, we could also compare dose-responses determined in this, and previous work performed with VSV.G-LV. We then compared GP64-LV and VSV.G-LV resistance to complement-mediated inactivation. Indeed, systemically administered LV reach the liver through the bloodstream, thus can be targeted by the complement system, an important component of innate immunity that acts as a first barrier both directly lysing vector particles and signaling as a pro-phagocytic factor for professional phagocytes in the liver and spleen. We have already reported that complement-mediated inactivation of VSV.G-LV in human serum is dose-dependent and is likely to be saturated at therapeutically relevant doses in vivo (Milani et al, 2017). Here, we incubated decreasing concentrations of GP64-LV or VSV.G-LV with complement-preserved, or heat-inactivated (complement-deprived) serum, and measured residual transduction of Huh7 cells, compared to LV not exposed to serum, set as 100% transduction. This previously reported in vitro assay reads out neutralization of transduction by complement-mediated lysis and/or binding by antibodies or other serum components (Milani et al, 2017). As the doses lowered, we observed decreased stability of both LV pseudotypes, independent of complement, i.e., heat-inactivated serum conditions (Fig. 1E–I). Whereas we confirmed

complement-mediated inactivation of VSV.G-LV at low doses, we observed comparable transduction by GP64-LV when incubated with heat-inactivated or complement-preserved serum, suggesting that complement does not play a role in the reduction of Huh7 transduction by GP64-LV (see Fig. 1E–I). These data confirmed previous reports of higher resistance of complement-mediated lysis by GP64-LV compared to VSV.G-LV. We went further and assessed the impact of complement-mediated opsonization of LV on phagocytosis mediated by primary human macrophages. To this aim, we set up an ad hoc in vitro LV phagocytosis assay.

## In vitro LV phagocytosis assay shows opsonization of GP64-LV

We produced fluorescent LV, carrying GFP fused to the membrane-targeting domain of pp60$^{Src}$, a chimeric protein previously shown to be effectively incorporated in the budding HIV envelope (Campbell et al, 2007; Milani et al, 2019; Rodgers, 2002). These fluorescent LV particles can be visualized post-entry in primary human macrophages using ImageStream, a combined flow cytometry and imaging system that allows high throughput quantification of LV entry (Fig. 2A). To assess the contribution of envelope-mediated surface binding to LV entry, compared to entry mediated by phagocytosis, we exploited LV lacking an envelope protein, thus unable to perform envelope-mediated entry (Bald-LV). We compared the distribution of GFP-positive spots in 293T cells and primary human macrophages after exposure to VSV.G-LV or Bald-LV. We observed more GFP spots in macrophages than 293T for both LV, confirming active capture of LV particles by macrophages (Fig. 2B), consistent with our previous report of higher transduction of macrophages compared to 293T by VSV.G-LV (Milani et al, 2019). These data show that the assay is fit for the purpose of assessing phagocytosis of LV by primary macrophages. By using this assay, we observed increased phagocytosis of VSV.G-LV when pre-incubated with a pool of sera obtained from healthy donors compared to pre-incubation with medium only, suggesting that serum components opsonize the LV particles, thus favoring their uptake by macrophages (Fig. 2C). Note that, at the LV doses used in the phagocytosis assay, there is no detectable direct complement-mediated LV inactivation. We have previously shown that modulating the levels of CD47 on LV particles affects their uptake by human macrophages (Milani et al, 2019). Since the i.v. administered LV is exposed to opsonizing antibodies (Abs) and complement, we assessed the role of CD47 in protecting opsonized LV from increased phagocytosis. We pre-incubated CD47hi fluorescent VSV.G-LV particles with complement-preserved human serum or medium, as control, and incubated the opsonized LV with primary human macrophages. We did not observe any increase in uptake by CD47hi LV even when opsonized (Fig. 2D). These data suggest that basal CD47 content is not enough to protect opsonized LV from phagocytosis in this experimental setting and that a high amount of CD47 on the LV surface effectively counterbalances the pro-phagocytic signals mediated by opsonization. We then evaluated phagocytosis of GP64-LV with or without pre-exposure to human serum and found detectable uptake by human macrophages, which was increased in the presence of human serum (Fig. 2E). Moreover, we observed a similar extent of phagocytosis of opsonized GP64-LV even when they were produced by vector-producer cells overexpressing CD47 (Fig. 2F). We thus analyzed CD47 incorporation into the envelope of GP64-LV produced by CD47-overexpressing cells, through immunostaining and electron microscopy and found a non-homogenous distribution of CD47 content among virions pseudotyped with GP64

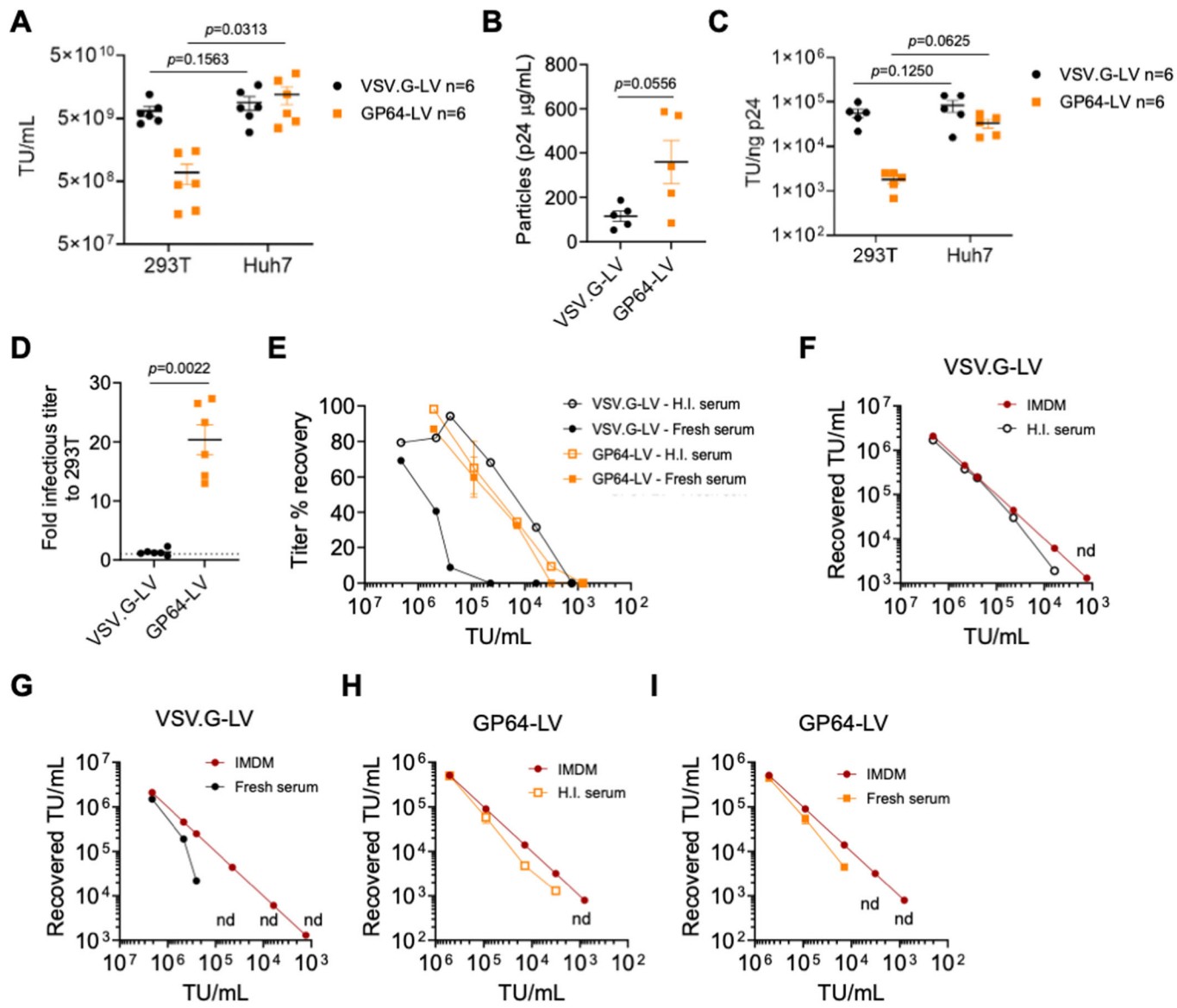

**Figure 1. In vitro characterization of GP64-pseudotyped LV.**

(A–C) Single values, mean, and standard deviation of the mean (SEM) of LV-infectious titer (A), physical particles (B) or specific infectivity (C) of VSV.G-LV or GP64-LV measured by transducing 293T or Huh7 cell lines, as indicated ($n = 6$ biological replicates). TU Transducing Units. (A, C) Wilcoxon matched-pairs signed rank test; (B) Mann–Whitney test. (D) GP64-LV-infectious titer measured on Huh7 is 20-fold higher compared to that measured on 293 T cells ($n = 6$), while VSV.G-LV transduce both cell lines similarly ($n = 6$). Single values, mean and SEM. Mann–Whitney test. (E). Percentage of titer recovered, compared to the no-serum control on Huh7 cell line (2 independent assays performed at the indicated LV concentration, $n = 1$ per dose per LV, except for $n = 2$ for GP64-LV at 9e4 TU/mL, mean and SEM) of GP64-LV or VSV.G-LV, incubated for 1 h with heat-inactivated (H.I., empty symbols) or fresh complement-preserved (filled symbols) human sera. (F–I) Recovered titer on Huh7 (y axis), of VSV.G-LV (F, G) or GP64-LV (H, I) as indicated, at the indicated concentration (x axis), incubated for 1 h in medium only (IMDM) or in H.I. (F, H) or fresh (G, I) serum. Data used to calculate percentages of titer recovered shown in (E). nd not detectable. Source data are available online for this figure.

protein, with some of them displaying similar amounts to control GP64-LV particles (Fig. EV1). Overall, these data show that GP64-LV is susceptible to macrophage-mediated phagocytosis.

## GP64-LV efficiently transduce mouse LSEC in vivo

We proceeded with the in vivo administration of GP64-LV to mice. We produced GP64-LV or VSV.G-LV expressing GFP under the control of the ubiquitously expressed phosphoglycerate kinase

(PGK) promoter, to assess transduction of different liver cell types. We administered GP64-LV.GFP or VSV.G-LV.GFP i.v. to wild-type (wt) C57BL/6 adult mice at increasing doses and determined LV copies per cell (vector copy number, VCN) and GFP-positive tissue area in the liver 1 week after LV administration. We observed an LV-dose-dependent increase of both liver VCN and GFP marking (Fig. 3A,B), except that VCN was similar in GP64-LV-treated mice at 2e10 or 4e10 TU/kg doses. A similar GFP-positive tissue area was found in GP64-LV and VSV.G-LV-treated mice.

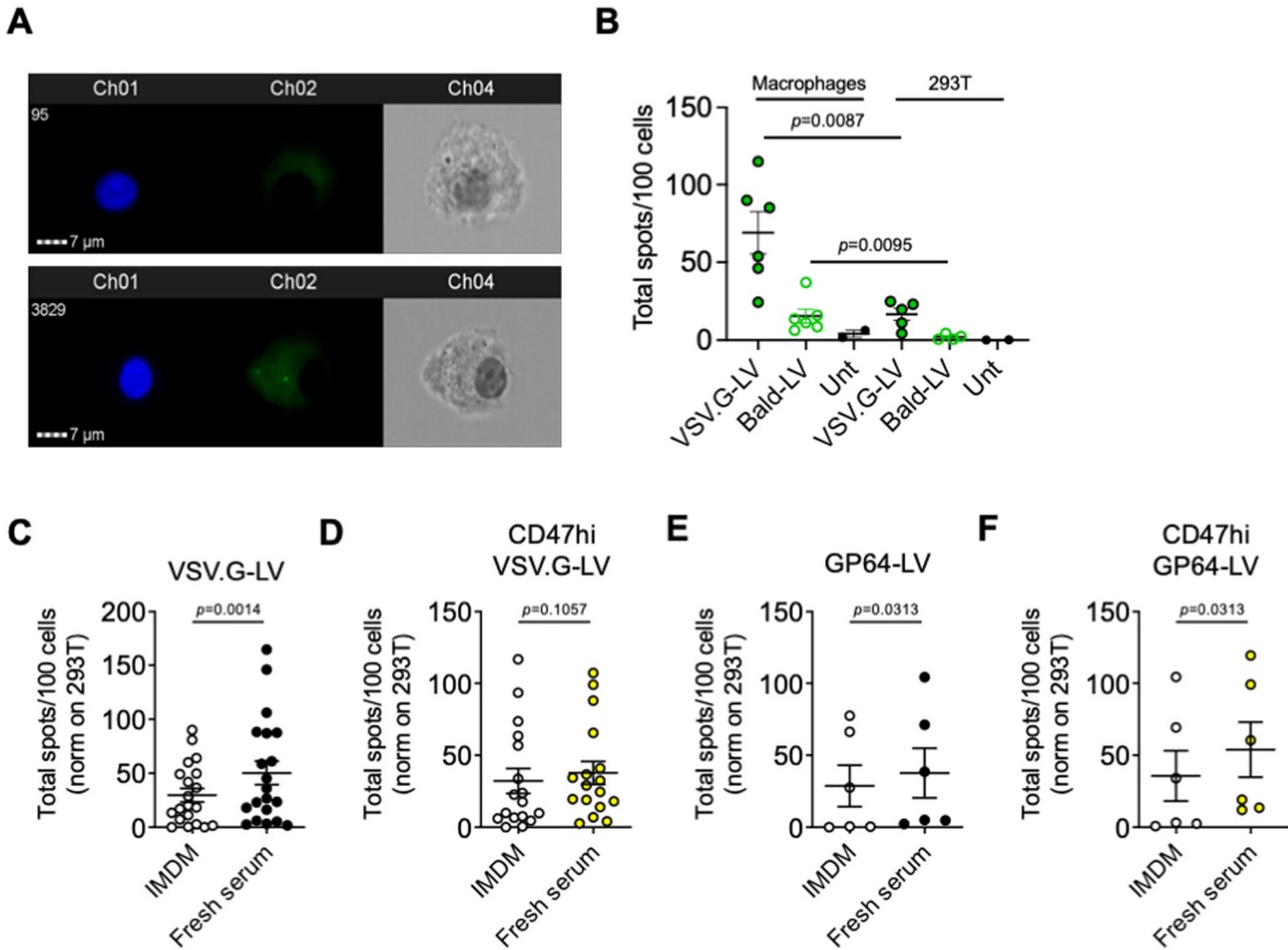

**Figure 2. Evaluation of phagocytosis of complement-opsonized LV.**

(A) Representative ImageStream images of macrophages left untreated (top panel) or incubated 6 h with fluorescent LV particles (bottom panel). Ch1: Hoechst nuclear staining; Ch2: GFP-positive LV clusters; Ch3: Brightfield. (B) Mean and SEM with single values of total LV spots per 100 primary human macrophages or 293T cells, as indicated, analyzed by ImageStream after incubation with VSV.G-LV, LV lacking envelope protein (Bald-LV) or left untreated (three independent experiments performed with macrophages derived from six different normal donors). Mann–Whitney test. (C–F) Mean and SEM with single values of phagocytosed VSV.G-LV (C, D) or GP64-LV (E, F) opsonized with fresh complement-preserved human sera (Fresh serum) compared to no-serum control (IMDM) by primary human macrophages analyzed by ImageStream. LV tested in these assays have been produced in standard 293 T cells (C, $n = 20$ normal donors; E, $n = 6$), or in CD47hi 293T cells (CD47hi LV, D, $n = 17$; F, $n = 6$). Multiplicity of infection (MOI) 10. Wilcoxon matched-pairs signed rank test. Source data are available online for this figure.

When we analyzed the distribution of the GFP signal among the main liver cell types, we observed that, at increasing doses, VSV.G-LV transduced more hepatocytes at the expense of KC, consistently with our previous report (Milani et al, 2019). On the other hand, high doses of GP64-LV resulted in increased transduction of LSEC, paralleled by a relative decrease in KC transduction (Figs. 3C,D and EV2). In particular, we detected a statistically significant difference in the percentage of LSEC among GFP-positive cells, at the highest LV dose tested (see Fig. EV2C). By looking at high-magnification immune fluorescence images, we confirmed that most GFP-positive cells in GP64-LV-treated mice were Lyve1-positive, while we observed both GFP-positive LSEC and KC in VSV.G-LV-treated mice. We then investigated the biodistribution of GP64-LV. We administered the same dose of GP64-LV.GFP or VSV.G-LV.GFP i.v. to adult C57BL/6 mice and determined the

GFP-positive area in different organs and the VCN in the liver 10 days post LV. We found that GFP marking was generally lower in mice treated with GP64-LV compared to those treated with VSV.G-LV (Fig. EV3A–E). The majority of GFP signal was coming from the liver and the spleen, as expected. Notably, except for the liver, almost no endothelial cells resulted GFP-positive in other organs, for both vectors, by morphology analysis (Fig. EV3F,G).

## CD47hi GP64-LV improve transduction of mouse LSEC in vivo

In parallel, we also produced GP64-LV or VSV.G-LV expressing human FIX under the control of the previously described hepatocyte-specific expression cassette (Milani et al, 2019), in order to more quantitatively compare transduction of hepatocytes between the two

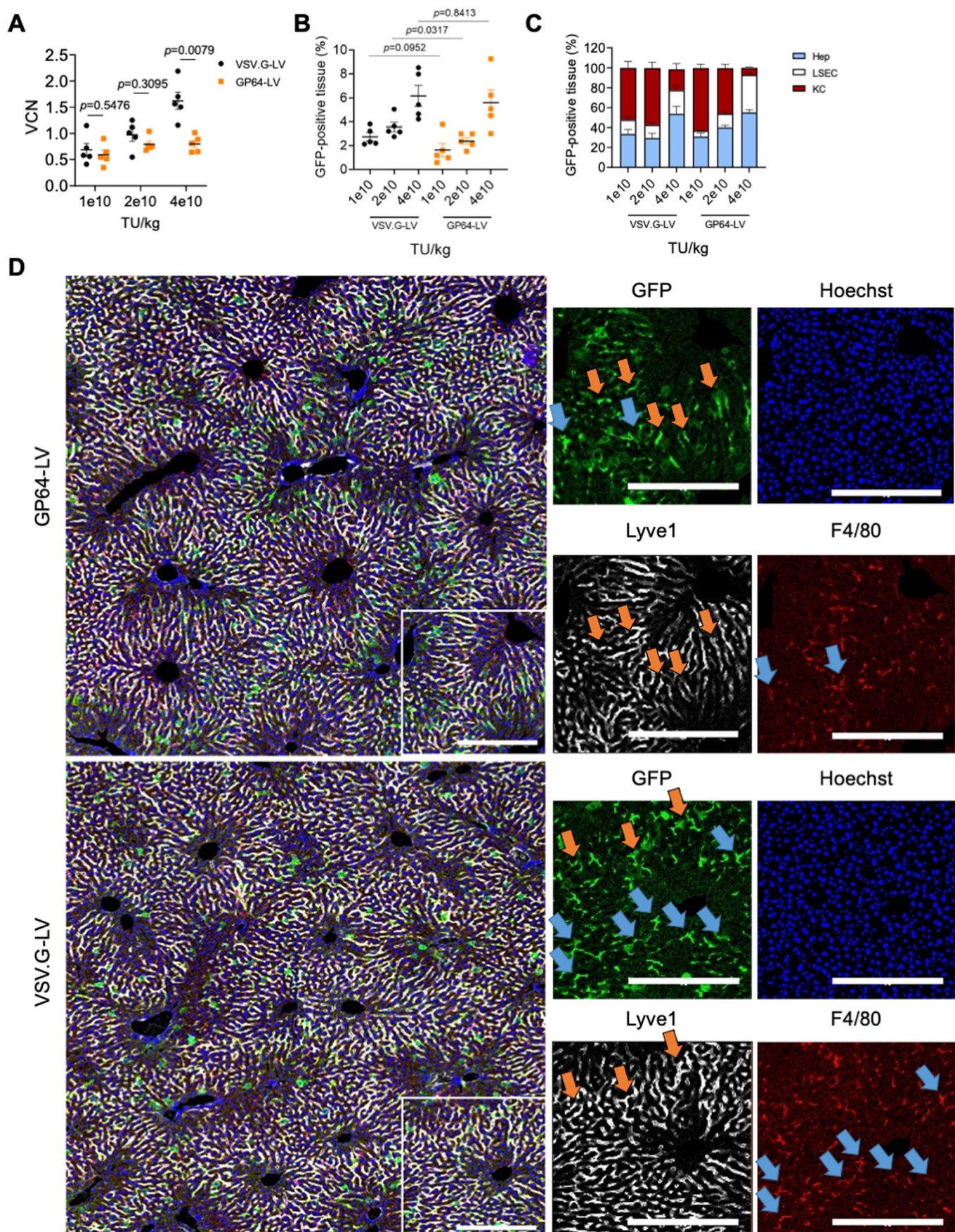

◄ **Figure 3. GP64-LV efficiently transduce liver sinusoidal endothelial cells in vivo.**

(A, B) Mean and SEM with single values of the VCN (A) or GFP-positive tissue (B) measured in the total liver of 8-week-old mice 1 week after i.v. administration of VSV.G-LV or GP64-LV at the indicated doses ($n = 5$ mice per group). Mann–Whitney test. (C) Mean and SEM of the distribution of the GFP-positive tissue among the major liver cell types of mice shown in (A, B) at the indicated doses (KC Kupffer cells, LSEC liver sinusoidal endothelial cells, Hep hepatocytes). (D) Representative images and merge of immunofluorescent staining of liver analyzed in B and C at 4e10 TU/kg ($n = 5$ mice). Red: F4/80-positive cells (KC); white: Lyve1-positive cells (LSEC); green: GFP (transduced cells); blue: Hoechst (nuclei). Orange arrows indicate GFP-positive LSEC. Blue arrows indicate GFP-positive KC. Scale bar: 250 μm. Source data are available online for this figure.

pseudotypes. We administered GP64-LV.FIX to adult mice, either C57BL/6 or of the NOD strain, which harbor a polymorphism in the CD47 receptor allowing cross-recognition of the human ligand, present on the LV particles (Milani et al, 2019; Takenaka et al, 2007). For this reason, NOD mice allow for assessing the impact of the presence of surface human CD47 on LV transduction. We observed higher circulating human FIX in NOD compared to C57BL/6 mice, at high GP64-LV doses (Fig. 4A). However, human FIX output in GP64-LV.FIX-treated C57BL/6 mice was substantially lower than that observed in VSV.G-LV-transduced C57BL/6 mice. These data suggest that, since FIX expression is driven by a hepatocyte-specific promoter, VSV.G-LV achieve higher transduction of hepatocytes than GP64-LV, besides similar GFP-positive hepatocyte area observed in the experiment shown above (see Figs. 3C and EV2B). We thus determined VCN from the main liver cell types upon FACS-sorting and found high VCN in LSEC and low VCN in hepatocytes (Fig. 4B). These data suggest that transduced LSEC contain multiple LV copies, while transduced hepatocytes likely carry a single LV copy. This molecular analysis provided additional insights into GP64-LV biodistribution among different liver cell types, complementing the results obtained with GFP (see Fig. 3C). We also found lower VCN in KC and plasmacytoid dendritic cells (pDC) in GP64-LV treated NOD compared to C57BL/6 mice, likely due to the sensing of the basal human CD47 present on LV particles in the NOD mouse strain, as we have previously shown for VSV.G-LV (see Fig. 4B). At the high LV dose, the VCN in both hepatocytes and LSEC increased, while it decreased in KC in NOD mice, suggesting that protection from uptake by professional phagocytes in vivo, mediated by the CD47, improved gene transfer into hepatocytes and LSEC by GP64-LV. These data prompted us to evaluate CD47hi GP64-LV in vivo. We administered CD47hi GP64-LV.FIX i.v. to adult NOD mice at increasing doses, then longitudinally measured circulating human FIX and determined VCN in liver cell subpopulations at the end of the experiment (3 months post LV). We observed a dose-dependent increase in FIX output mediated by the CD47hi GP64-LV.FIX in NOD mice (Fig. 4C), which was similar to that achieved by GP64-LV.FIX in NOD mice (see Fig. 4A), consistently with the hepatocyte-derived FIX production. We also measured circulating LV particles soon after administration and found that half-life was prolonged in NOD mice compared to C57BL/6, likely due to reduced phagocytosis in the former mouse strain, by sensing of human CD47 signal on LV particles. Accordingly, the longest half-life was obtained when administering CD47hi GP64-LV to NOD mice (Fig. EV4A), in agreement with previous observations with VSV.G-LV (Milani et al, 2019). We did not observe an increase in the VCN of hepatocytes, by using CD47hi GP64-LV at the highest dose, and VCN in hepatocytes was in line with FIX output, which reached 20% of normal. In contrast, we found high VCN in LSEC, reaching >10, which was even higher than that observed in GP64-LV.FIX-treated NOD mice (Fig. 4D,E). We observed slightly lower

transduction of a hepatocyte cell line by CD47hi GP64-LV compared to GP64-LV (Fig. EV4B), suggesting that the increased transduction of LSEC in vivo was not due to a generally higher transduction efficiency of CD47hi GP64-LV, but rather to higher protection from uptake by professional phagocytes in vivo and subsequent higher availability for transduction of LSEC. Overall, these data confirm a high efficiency of gene transfer to LSEC in vivo mediated by GP64-LV systemic administration, further increased by exploiting CD47hi LV.

## GP64-LV efficiently transduce primary human LSEC

We then set out to compare the transduction of primary human hepatocytes or LSEC, by GP64-LV or VSV.G-LV expressing GFP, FIX or a human codon-optimized FVIII transgene (Milani et al, 2022), under the control of its endogenous endothelial-specific promoter, previously described (Fama et al, 2021). We transduced primary human hepatocytes at different multiplicities of infection (MOI) with GFP or FIX-expressing vectors. In line with the above-mentioned results obtained in vivo, we found higher transduction efficiency by VSV.G-LV than GP64-LV at all tested MOI, as assessed by determining GFP-positive cells and intensity of the GFP signal within the positive cells, or by measuring the FIX present in the supernatant of transduced cells, and by quantifying VCN (Fig. 5A–D). Similarly, also primary hepatocytes from different mouse strains were transduced less efficiently by GP64-LV than VSV.G-LV ex vivo in all tested conditions (Appendix Fig. S1). We then transduced primary human immortalized LSEC (Appendix Fig. S2) with GP64-LV expressing GFP of FVIII. Notably, the transduction efficiency of these LSEC was similar or even higher by GP64-LV than VSV.G-LV, particularly at low MOI and also FVIII production measured in the cells' supernatant was higher in GP64-LV transduced cells than in VSV.G-LV transduced cells (Fig. 5E–H). To evaluate the transduction of other endothelial cells different from LSEC, we transduced primary human blood outgrown endothelial cells (BOEC) from a human donor affected by hemophilia A. We observed higher efficiency of transduction by VSV.G-LV compared to GP64-LV at all tested MOI (Fig. EV5). Overall, these data show lower efficiency of transduction of primary human hepatocytes and vascular endothelial cells by GP64-LV compared to VSV.G-LV and vice versa for primary human LSEC. These data are in line with the mouse biodistribution data shown above (see Fig. EV3) and suggest that highly efficient gene transfer by GP64-LV is specifically achieved in LSEC.

## GP64-LV achieve therapeutically relevant FVIII expression mediated by the endogenous promoter in hemophilia A mice

The marked in vivo tropism for LSEC by GP64-LV prompted us to evaluate FVIII gene transfer in hemophilia A mice. We thus treated

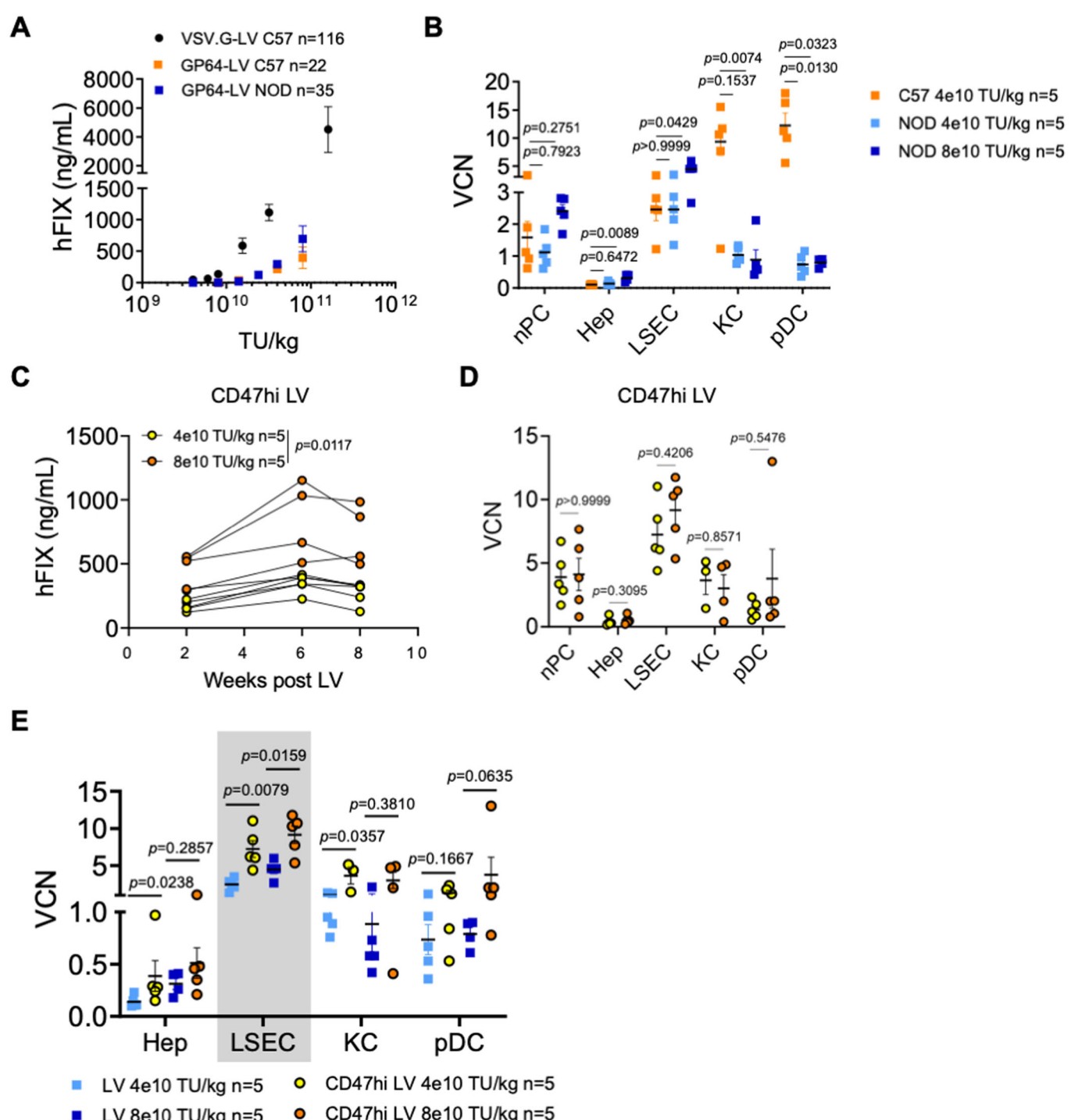

**Figure 4.  In vivo evaluation of CD47hi GP64-LV.**

(A) Mean with SEM of human FIX (hFIX) concentration measured in the plasma of C57BL/6 mice or NOD mice, as indicated, treated at the indicated LV doses with VSV.G-LV ($n = 116$, historical dataset from twenty-four independent experiments) or GP64-LV ($n = 22$ C57BL/6, $n = 35$ NOD, from four independent experiments, performed with four different LV batches). (B) Mean with SEM with single values of VCN in measured in non-parenchymal cells (nPC), FACS-sorted hepatocytes (Hep), LSEC, KC, and plasmacytoid dendritic cells (pDC) as indicated, of C57BL/6 mice ($n = 5$) or NOD mice ($n = 5$), 2 months after administration of the indicated doses of LV. Kruskal–Wallis test with Dunn's multiple comparisons test (compared to C57BL/6 mice). (C) Single values of human FIX (hFIX) concentration measured over time in the plasma of NOD mice treated at the indicated LV doses with CD47hi GP64-LV ($n = 5$). Two-way ANOVA. (D) Mean with SEM with single values of VCN measured in liver subpopulations as indicated, of mice shown in (A), 2 months after administration. Mann–Whitney test. (E) Mean with SEM with single values of VCN measured in the indicated liver subpopulations of mice shown in Fig. 3F (LV) and 4B (CD47hi LV), reported here for direct comparison. Mann–Whitney test. Source data are available online for this figure.

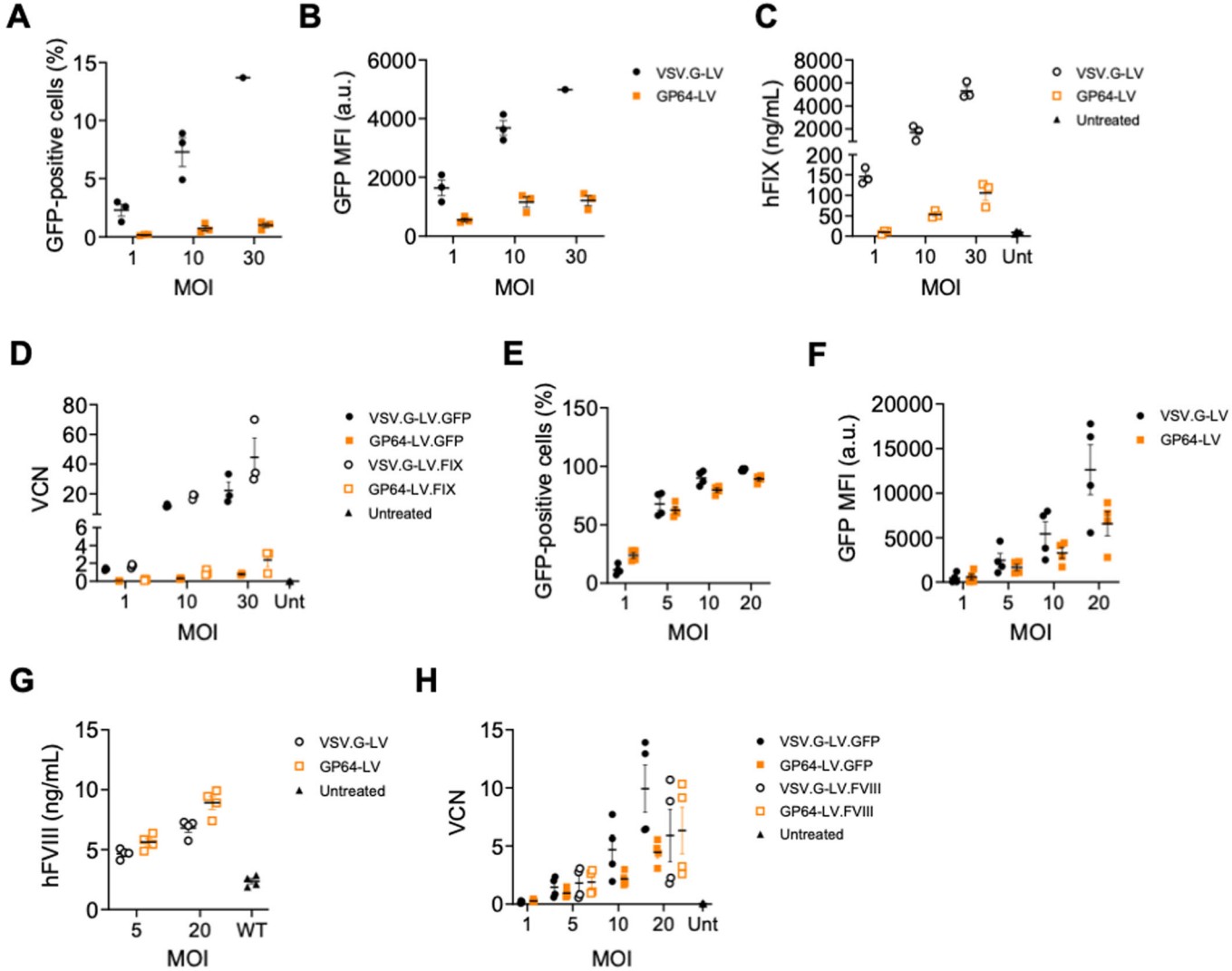

**Figure 5. Ex vivo transduction of primary human hepatocytes and immortalized LSEC.**

(A, B) Mean and SEM with single values of the % (A) or mean fluorescent intensity (MFI, B) of GFP-positive primary human hepatocytes (*n* = 3 technical replicates) transduced with VSV.G-LV or GP64-LV at the indicated multiplicity of infections (MOI). Mann–Whitney test. (C) Mean and SEM with single values of the amount of hFIX measured in the culture supernatant of primary human hepatocytes (*n* = 3 technical replicates) transduced with VSV.G-LV or GP64-LV at the indicated MOI, or left untreated. (D). Mean and SEM with single values of the vector copy numbers (VCN) of reversed transcribed LV genome of the primary human hepatocytes (shown in (A–C), *n* = 3 technical replicates) transduced with VSV.G-LV or GP64-LV encoding for GFP or hFIX at the indicated MOI, or left untreated. (E, F) Mean and SEM with single values of the % (E) or MFI (F) of GFP-positive primary human immortalized LSEC derived from a healthy donor (*n* = 4 independent experiments) transduced with VSV.G-LV or GP64-LV at the indicated MOI. (G) Mean and SEM with single values of the amount of hFVIII measured in the culture supernatant of primary human immortalized LSEC derived from a healthy donor (*n* = 4 independent experiments) transduced with VSV.G-LV or GP64-LV at the indicated MOI, or left untreated. (H) Mean and SEM with single values of the VCN of reversed transcribed LV genome of the primary human LSEC (shown in (E–G)) transduced with VSV.G-LV or GP64-LV encoding for GFP or hFVIII at the indicated MOI, or left untreated. Source data are available online for this figure.

adult immune tolerant hemophilia A mice (*F8* knock out, KO, expressing a human mutated FVIII, R593C) (Bril et al, 2006) by i.v. delivery of the same dose of GP64-LV or VSV.G-LV expressing the codon-optimized human FVIII under the control of the *F8* promoter (Merlin et al, 2019). We longitudinally monitored circulating human FVIII expression and activity in treated mice. Interestingly, treatment with GP64-LV.FVIII allowed reconstituting about 10–20% of normal FVIII antigen and activity, while FVIII was undetectable in mice treated with VSV.G-LV.FVIII at this LV dose (Fig. 6A,B). Note that the FVIII expression cassette used in the

present study is different from that used by Merlin et al, in a previous report (Merlin et al, 2017). Anti-human FVIII Abs were lower in GP64-LV treated mice (Fig. 6C). We determined the FVIII neutralizing activity in the plasma of treated mice and showed that only 1/11 treated mice had a transient neutralizing Ab titer, spontaneously disappearing at later times (Fig. 6D). These data suggest that immune response is limited to the generation of binding but not neutralizing anti-FVIII Abs, without precluding FVIII persistence. Mice were then subjected to a hemostasis challenge to test phenotypic correction. All mice treated with

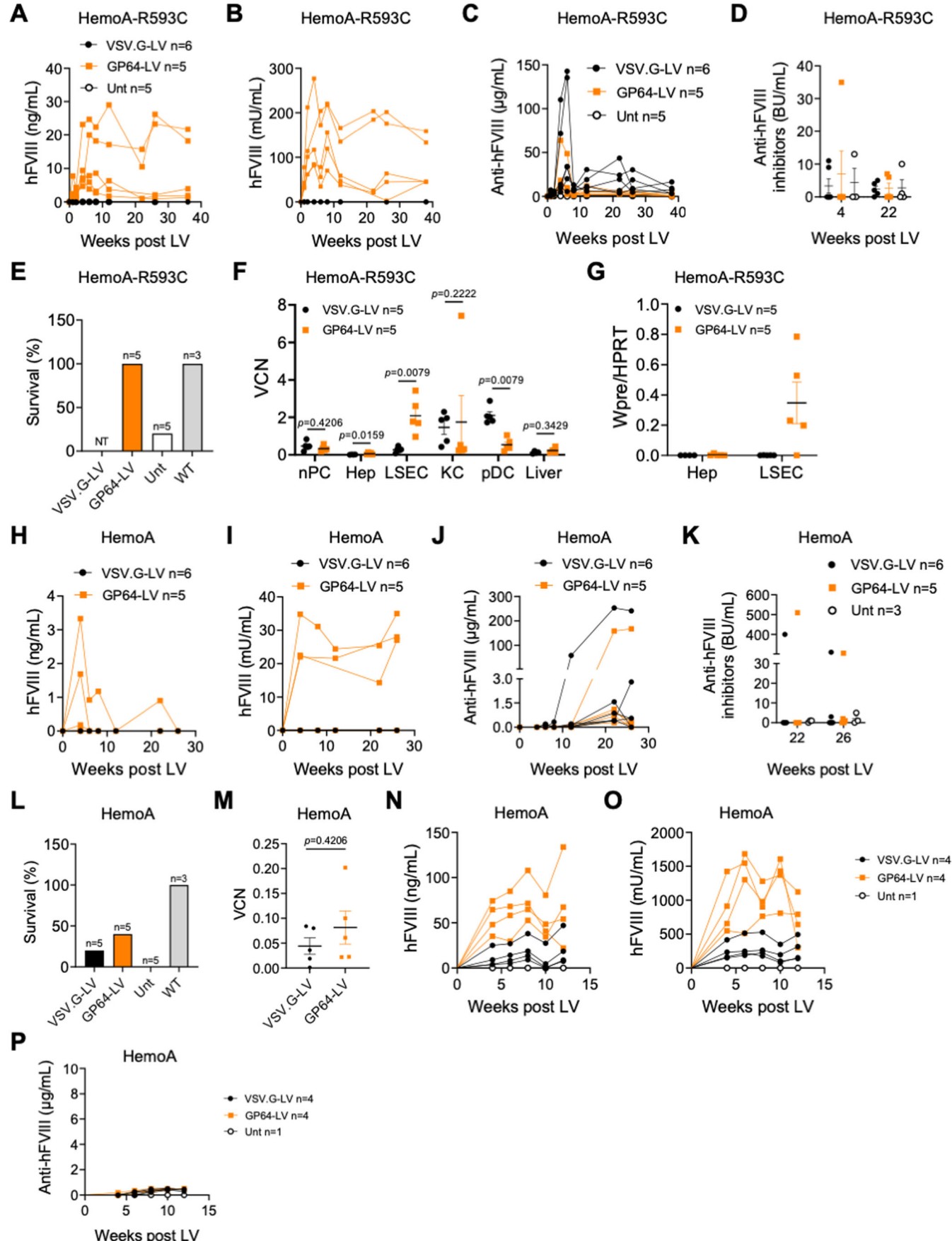

**Figure 6. Evaluation of GP64-LV mediated gene therapy in hemophilia A mice.**

(A–C) Single values of hFVIII antigen (A), activity (B), or anti-hFVIII antibodies (Abs, C) measured over time in the plasma of immune-competent HemoA-R593C mice treated at 7 weeks of age by i.v. injection of $5 \times 10^{10}$ TU/kg of VSV.G-LV ($n = 6$) or GP64-LV ($n = 5$) of LV-F8.coFVIII, or left untreated ($n = 5$). (D) Single values and mean with SEM of anti-hFVIII inhibitors of mice shown in (A–C) at the indicated time after LV administration. BU Bethesda Units. (E) Survival of mice shown in (A–C) 24 h after hemostatic challenge. Note that mice treated with VSV.G-LV were not tested (NT) since they do not display measurable hFVIII in the circulation. WT wild type. (F) Mean with SEM with single values of VCN measured in nPC, FACS-sorted Hep, LSEC, KC, pDC and whole liver, as indicated, of mice shown in (A–C), 36 weeks after LV administration. Mann–Whitney test. (G) Mean with SEM with single values of LV expression measured in FACS-sorted Hep or LSEC, as indicated, of mice shown in (A–C), 36 weeks after LV administration. WPRE: woodchuck hepatitis B virus post-transcriptional regulatory element, present in the transgene mRNA. (H–J) Single values of hFVIII antigen (H), activity (I) or anti-hFVIII Abs (J) measured over time in the plasma of immune-competent HemoA mice treated as newborns by temporal vein injection of $3 \times 10^{10}$ TU/kg of VSV.G-LV ($n = 6$) or GP64-LV ($n = 5$) of LV-F8.coFVIII. (K) Single values and mean with SEM of anti-hFVIII inhibitors of mice shown in (H–J) at the indicated time after LV administration. (L) Survival of mice shown in (F–H) 24 h after hemostatic challenge. WT wild type. (M) Mean with SEM with single values of VCN measured in the total liver of mice shown in (F–H), 36 weeks after LV administration. Mann–Whitney test. (N–P) Single values of hFVIII antigen (N), activity (O) or anti-hFVIII Abs (P) measured over time in the plasma of immune-competent HemoA mice treated as newborns by temporal vein injection of $9 \times 10^{10}$ TU/kg of VSV.G-LV ($n = 4$) or GP64-LV ($n = 4$) of LV-F8.coFVIII. Source data are available online for this figure.

GP64-LV survived, as well as wild-type mice, indicating hemostasis restoration in the former, while all control untreated mice, except for one, died. VSV.G-LV-treated mice were not challenged, for ethical reasons, since they did not display detectable amounts of human FVIII in the circulation (Fig. 6E). Interestingly, VCN in LSEC was higher in GP64-LV than VSV.G-LV-treated mice (Fig. 6F). Moreover, we only detected LV transgene mRNA in LSEC of GP64-LV but not in VSV.G-LV-treated mice, in line with the data of FVIII protein. We did not detect LV transgene mRNA in purified hepatocytes, confirming endothelial-cell-specific expression of the F8 promoter (Fig. 6G). In order to assess the stability of LSEC-derived FVIII following liver growth, we administered GP64-LV.FVIII or VSV.G-LV.FVIII to newborn immune-competent hemophilia A mice (F8 KO). We achieved 2–3% of human FVIII expression and activity in 3/5 GP64-LV treated mice, which remained stable for up to 26 weeks post LV (Fig. 6H,I). Some of the mice developed neutralizing anti-FVIII Abs at later times after LV administration, as previously shown with VSV.G-LV expressing FVIII transgene by hepatocytes (Fig. 6J,K; (Milani et al, 2022)). Despite the low amount of human FVIII in the circulation, 2/5 GP64-LV-treated mice with detectable FVIII survived the hemostasis challenge, while only one mouse treated with VSV.G-LV survived (Fig. 6L). VCN in total liver was comparable between VSV.G- and GP64-LV-treated mice (Fig. 6M). We treated additional newborn hemophilia A mice with VSV.G-LV or GP64-LV expressing FVIII at threefold higher vector doses. Remarkably, we achieved around 100% of normal FVIII activity in GP64-LV-treated mice which was fourfold higher than in VSV.G-LV-treated mice (Fig. 6N,O). We did not detect anti-FVIII Abs (Fig. 6P). These data confirm the higher efficiency of gene transfer into LSEC by GP64-LV compared to VSV.G-LV and highlight the possible use of GP64-LV for in vivo LSEC-mediated hemophilia A gene therapy.

## Discussion

Here we show that GP64-LV is suitable for in vivo gene transfer since it is resistant to complement-mediated inactivation, allows highly efficient targeting of LSEC, and thus can be exploited to express FVIII in its physiologic site of production, achieving correction of hemophilia A in mice. Effective modulation of viral vector tropism is an important goal for gene therapy, to improve its specificity and overall efficacy and safety. When comparing vectors with different envelope proteins, it is crucial to identify an appropriate method to match doses, either based on the measurement of physical vector particles or by determining infectious titers in equally permissive cell lines. Assessment of TU in cell lines of different permissiveness to the differently pseudotyped vectors would otherwise result in incorrect estimation of the administered dose (e.g., apparently similar TU corresponding to markedly different physical particles) and confound the interpretation of the results. For this reason, here we identified the Huh7 as a suitable cell line for this purpose. We previously exploited primary human macrophage cultures as a read-out of vector susceptibility to phagocytosis. Here, we improved our assessment, by evaluating the contribution of pro- and anti-phagocytic signals to this outcome. Indeed, we developed a novel in vitro assay to quantify phagocytosis of LV exploiting fluorescent viral particles, primary human macrophages, and an advanced single-cell level imaging technique. This assay is useful to assess the susceptibility to phagocytosis of different versions of LV with surface modifications, in the presence or absence of opsonizing signals and/or of molecules interfering with complement activation or the phagocytosis process. We found that GP64-LV are compatible with the incorporation of high amounts of human CD47 when produced by cells overexpressing it, as previously shown for VSV.G-LV. However, contrarily to CD47hi VSV.G-LV that were uniformly CD47hi, we noted that a fraction of GP64-pseudotyped virions display the basal amount of CD47, thus suggesting that changing the LV envelope protein can affect the global protein composition of the vector surface. Indeed, we observed a different outcome of CD47hi VSV.G-LV compared to GP64-LV both in vitro and in vivo. Specifically, CD47hi VSV.G-LV appeared consistently protected from phagocytosis even when opsonized by human serum in the in vitro assay. VSV.G-LV showed reduced uptake by KC in NOD mice, as previously shown. Instead, CD47hi GP64-LV were phagocytosed similarly to GP64-LV with basal CD47 content in vitro. This may be due to the non-homogenous incorporation of CD47 on GP64-LV particles. Despite this observation, CD47hi GP64-LV transduced LSEC in vivo at very high efficiency, particularly at high vector doses. On the other hand, even at high GP64-LV doses, transduction of hepatocytes remains limited, and hepatocyte-derived FIX output never matched that achieved by VSV.G-LV. Molecular analysis of VCN in different liver cell types provided additional insights, complementing the analysis of GFP-positive liver cells. A similar GFP-positive area may correspond to cells carrying different numbers of LV copies, thus VCN data

suggest that the LSEC transduced by GP64-LV have multiple LV copies per cell, while transduced hepatocytes likely carry a single LV copy. This is particularly relevant for secreted transgenes such as clotting factors, since a cell transduced at multiple LV copies results in higher transgene output in the bloodstream compared to a cell transduced with a single LV copy. In line with the higher FIX in vivo output, gene transfer into primary human hepatocytes was also more efficient with VSV.G-LV than GP64-LV. Similar results were observed when comparing the transduction efficiency of primary human BOEC by VSV.G-LV or GP64-LV, with the former resulting in better gene transfer also in this cell type. These data suggest that improved in vivo tropism for LSEC of GP64-LV compared to VSV.G-LV is the result of high bioavailability in the liver microvasculature coupled with low efficiency of hepatocyte transduction. In addition, GP64-LV appeared quite selective for LSEC, since, in contrast to hepatocytes and BOEC that were better transduced by VSV.G-LV than GP64-LV, human LSEC were similarly or better transduced by the latter than the former. This feature makes GP64-LV attractive vehicles for the delivery of FVIII transgene. Indeed, when equipped with a FVIII expression cassette driven by a reconstructed endothelial-specific *F8* promoter, GP64-LV improved therapeutic efficacy in hemophilia A mice compared to VSV.G-LV, achieving reconstitution of therapeutically relevant FVIII activity up to the normal range and correction of the bleeding phenotype in hemophilia A mice. Interestingly, GP64-LV-mediated LSEC-derived FVIII output remained stable throughout the long-term follow-up in both mice treated when newborn or when adults, suggesting maintenance of the transduced LSEC following liver growth and homeostatic renewal. Of note, we have previously reported that hepatocyte-mediated FVIII production by i.v. delivery of VSV.G-LV achieved stable normal to supranormal FVIII activity in hemophilia A mice and NHP. It has been proposed that the expression of high amounts of FVIII by hepatocytes may cause cellular stress and toxicity (Kapelanski-Lamoureux et al, 2022; Poothong et al, 2020), while it may not be the case when physiological amounts are produced (Lange et al, 2016; Zolotukhin et al, 2016). To ultimately establish whether it would be more beneficial to express FVIII by hepatocytes or LSEC in gene therapy will require further investigation about efficacy, safety, durability, and immunogenicity. Overall, the results reported here show that GP64-LV is an attractive tool for in vivo gene transfer into LSEC with potential application in gene therapy for hemophilia A or other diseases that can benefit from LSEC-derived transgene expression.

# Methods

### Study design

Sample size in experiments with mice was chosen according to previous experience with experimental models and assays. No sample or animal was excluded from the analyses. Mice were randomly assigned to each experimental group. Investigators were not blinded.

### Plasmid construction

Plasmid pCCLsin.cPPT.ET.coFIX-R338L.142T and pCCLsin.cPPT.PGK.GFP were previously described. Packaging plasmid encoding for GP64 was previously described (Schauber et al, 2004). Plasmid pCCLsin.cPPT.F8.coFVIII.142T was constructed by standard cloning techniques, by exchanging the gene synthesized human FVIII cDNA (Milani et al, 2022) with FIX into the previously described pCCLsin.cPPT.ET.canineFIX.142T (NheI-SalI) and then exchanging the ET promoter with the *F8* promoter (Fama et al, 2021). GFP-pp60Src fusion construct was previously described (Milani et al, 2019).

### Vector production

Third-generation self-inactivating (SIN) LV were produced by calcium phosphate transient transfection into 293T cells. 293T cells were transfected with a solution containing a mix of the selected LV genome transfer plasmid, the packaging plasmids pMDLg/pRRE and pCMV.REV, pMD2.G or pBA-AcMNPV-gp64 (Schauber et al, 2004) and pAdVantage (Promega), as previously described (Milani et al, 2019). Briefly, medium was changed 14–16 h after transfection, and the supernatant was collected 30 h after medium change. LV-containing supernatants were sterilized through a 0.22-µm filter (Millipore), transferred into sterile poliallomer tubes (Beckman) and centrifuged at $20,000 \times g$ for 120 min at 20 °C (Beckman Optima XL-100K Ultracentrifuge). For FVIII transgene encoding LV, before centrifugation, LV-containing 0.22-µm-filtered supernatants were pre-concentrated four times with Vivaflow 200 Tangential Flow Filtration Cassette 100 kDa (Sartorius), according to manufacturing instructions. LV pellet was dissolved in the appropriate volume of PBS to allow 500–1000× concentration. To produce fluorescent virions, 15 µg/plate of pMAX.GFP$^{pp60Src}$ plasmid were added to plasmid mix in the absence of genome-encoding plasmid to produce empty LV particles. Bald-LV were produced in the absence of envelope-encoding packaging plasmid.

### LV titration

For LV titration, $1 \times 10^5$ 293T or Huh7 cells were transduced with serial LV dilutions in the presence of polybrene (8 µg/ml). For LV-GFP, cells were analyzed by flow cytometry using a FACSCanto analyzer (BD Biosciences), equipped with DIVA Software, 5–7 days after transduction, and infectious titer, expressed as transducing units (TU)/mL, was calculated using the formula TU/mL = ((% GFP+ cells/100) × 100,000 × (1/dilution factor)). For all other LV, genomic DNA Genomic DNA (gDNA) was extracted 14 days after transduction, using Maxwell 16 Cell DNA Purification Kit (Promega), following the manufacturer's instructions. VCN was determined was determined by ddPCR, starting from 5–20 ng of template gDNA using primers (HIV fw: 5'-TACTGACGCTCTCGC ACC-3'; HIV rv: 5'-TCTCGACGCAGGACTCG-3') and a probe (FAM 5'-ATCTCTCTCCTTCTAGCCTC-3') designed on the primer-binding site region of LV. The amount of endogenous DNA was quantified by a primers/probe set designed on the human telomerase gene (Telo fw: 5'-GGCACACGTGGCTTTTCG-3'; Telo rv: 5'-GGTGAACCTCGTAAGTTTATGCAA-3'; Telo probe: VIC 5'-TCAGGACGTCGAGTGGACACGGTG-3' TAMRA) or the human GAPDH gene (Applied Biosystems HS00483111_cm). The PCR reaction was performed with each primer (900 nM) and the probe (250 nM, 500 nM for Telo) following the manufacturer's instructions (Biorad), read with QX200 reader and analyzed with QuantaSoft software (Biorad). Infectious titer, expressed as TU/mL,

was calculated using the formula TU/mL = (VCNx100,000x(1/dilution factor). LV physical particles were measured by HIV-1 Gag p24 antigen immunocapture assay (Perkin Elmer) following the manufacturer's instructions. LV-specific infectivity was calculated as the ratio between infectious titer and physical particles.

## VCN determination

For experiment with primary hepatocytes, DNA was extracted using Maxwell 16 Tissue DNA Purification Kit (Promega), following the manufacturer's instructions. For mice experiments, DNA was extracted from whole-liver or whole-spleen samples using Maxwell 16 Tissue DNA Purification Kit (Promega), DNA was extracted from fractionated/sorted liver cells using DNeasy Blood & Tissue Kit (Qiagen) or QIAamp DNA Micro Kit (Qiagen), according to cell number. VCN was determined in Huh7 samples as described above (see "LV titration"). VCN in murine DNA line was determined by ddPCR, starting from 5 to 20 ng of template gDNA using a primers/probe set designed on the primer-binding site region of LV (see "LV titration" above). For primary hepatocytes, because cells do not divide in culture, VCN was determined using an ad hoc ddPCR (QX200 EvaGreen Digital PCR Supermix, Bio-Rad), which selectively amplifies the reverse transcribed vector genome (both integrated and non-integrated), discriminating it from plasmid carried over from the transient transfection (RT-LV; ΔU3 fw: 5'-TCACTCCCAACGAAGACAAGATC-3', gag rv: 5'-GAGTCCTGCGTCGAGAGAG-3') (Matrai et al, 2011). The amount of endogenous DNA was quantified by primers set designed on the human telomerase gene, as above, or the murine Sema3A gene, as described below. The amount of endogenous murine DNA was quantified by a primers/probe set designed on the murine sema3a gene (Sema3A fw: 5'-ACCGATTCCAGATGATT GGC-3'; Sema3A rv: 5'-TCCATATTAATGCAGTGCTTGC-3'; Sema3A probe: HEX 5'-AGAGGCCTGTCCTGCAGCTCATGG-3' BHQ1). The PCR reaction was performed with each primer (900 nM, 150 nM for RT-LV primers) and the probe (250 nM) following the manufacturer's instructions (Biorad), read with QX200 reader and analyzed with QuantaSoft software (Biorad).

## Cell cultures and in vitro experiments

293T and Huh7 cell lines were maintained in Iscove's modified Dulbecco's medium (IMDM, Sigma) supplemented with 10% fetal bovine serum (FBS, FetalClone II, HyClone, Euroclone), 4 mM glutamine (Lonza), penicillin and streptomycin 100 international units (IU)/mL (Lonza). Primary human monocytes were magnetically sorted (Miltenyi Biotec) from buffy coats of healthy donors. Primary human macrophages were obtained from CD14-positive cells isolated by negative selection (Pan Monocyte Isolation Kit, Miltenyi Biotec), from buffy coats of healthy donors (obtained according to a protocol approved by the S.R.S.I. Ethical Committee) and differentiated in IMDM, supplemented with 5% human serum, 4 mM glutamine, penicillin, and streptomycin 100 IU/mL for 7 days. The purity of CD14-positive cells was determined by flow cytometry and was >90%. All cells were maintained in a 5% $CO_2$ humidified atmosphere at 37 °C. All cell lines were routinely tested for mycoplasma contamination. Primary human hepatocytes were purchased from Biopredic and cultured following the manufacturer's instructions. Primary murine hepatocytes were obtained

from C57BL/6, NOD or *F8* KO hemophilia A mice as described below (see "Fractionation and sorting of liver cell sub-populations") and maintained in William's medium supplemented with penicillin and streptomycin 100 IU/mL (Lonza), sodium pyruvate (Gibco), non-essential amino acids (Gibco), 10% FBS (Euroclone), bovine serum albumin (Sigma, 1 mg/mL), ascorbic acid (Sigma, 250 µg/mL), insulin (Lonza, 5 µg/mL), olo-transferrin (Sigma, 10 µg/mL), hydrocortisone (Sigma, 500 ng/mL) and murine epidermal growth factor (R&D systems, 250 ng/mL). Primary murine or human hepatocytes were transduced for 24 h and then cultured for 3 days before gDNA extraction and VCN determination (see "VCN determination"). At this time, primary human hepatocytes transduced with a PGK.GFP LV were also collected for flow cytometry analysis to determine GFP expression using a FACS-Canto analyzer (BD Biosciences), equipped with DIVA Software, while in the case of primary human hepatocytes transduced with a ET.FIX.142mirT LV FIX-containing supernatant was collected, and hFIX concentration was assessed by ELISA (see "FIX assays"). Blood outgrowth endothelial cells (BOEC) were isolated from a healthy donor or a severe hemophilic patient (HA BOEC) with an intron 22 inversion mutation of the *F8* gene (protocol 1103/CE, study number CE 234/20). Cells were cultured on Collagen Type 1-coated flasks (Greiner Bio-One) using MCDB 131 medium (Gibco, Life Technologies) containing proprietary supplements (Olgasi et al, 2021). HA BOEC were plated at a density of $10^4$ cells/cm² and transduced after 8 h. Fresh medium was added to the cells after 16 h. BOEC transduced with a PGK.GFP LV were collected for flow cytometry analysis 72 h after transduction to determine GFP expression using a FACSCanto analyzer (BD Biosciences), equipped with DIVA Software, while in the case of BOEC transduced with a F8.FVIII LV FVIII-containing supernatant was collected and hFVIII concentration was assessed by ELISA (see "FVIII assays"). The supernatant was collected twice every 72 h. BOEC were cultured for 7 days before gDNA extraction and VCN determination (see "VCN determination"). Immortalized human LSEC were purchased from Innoprot (P10652-IM). Cells were cultured on Collagen Type 1-coated flasks (Greiner Bio-One) using MCDB 131 medium (Gibco, Life Technologies) containing supplements, as described (Olgasi et al, 2021). Cells were plated at a density of $10^4$ cells/cm² and transduced after 8 h. Fresh medium was added to the cells after 16 h. LSEC transduced with LV.GFP at MOI of 1, 5, and 10, were collected for flow cytometry analysis 72 h after transduction to determine GFP expression using a FACS-Canto analyzer (BD Biosciences), equipped with DIVA Software. For cells transduced with LV.FVIII, MOI of 5 and 10 were used and FVIII-containing supernatants were collected after 72 h and FVIII concentration was assessed by ELISA (see "FVIII assays"). LSEC were cultured for 10 days before gDNA isolation and VCN determination.

## RNA extraction and ddPCR

RNA extraction was performed using RNeasy Plus Micro Kit (Qiagen), according to the manufacturer's instructions and reverse transcribed using the SuperScript IV VILO kit (11766050; Thermo Fisher Scientific). LV gene expression was assessed by ddPCR starting from 10 to 25 ng of template cDNA using a primers/probe set designed on the woodchuck hepatitis B virus post-transcriptional regulatory element (WPRE) region of LV (WPRE:

primer fw 5'-GGCTGTTGGGCACTGACAAT-3'; primer rv 5'-ACGTCCCGCGCAGAATC-3'; probe FAM 5'-TTTCCTTGGCTGCTCGCCTGTGT-3' NGB). HPRT was used as a reference gene (dMmuCPE5095493, Biorad). The PCR reaction was performed with each primer (900 nM) and the probe (250 nM) following the manufacturer's instructions (Biorad), read with QX200 reader and analyzed with QuantaSoft software (Biorad).

## LV inactivation assay

Complement-preserved human serum was purchased (Sigma). Serum was thawed and half of each serum sample was heated at 56 °C for 1 h to inactivate the complement. LV were diluted in IMDM (Sigma) supplemented with 10% FBS (Euroclone). Twenty µL of LV were diluted 1:5 into fresh or heat-inactivated serum (or IMDM 10% FBS as the no-serum control) and the mixture was incubated at 37 °C for 1 h. Following incubation, medium was added to the reaction and then serially diluted and used to transduce Huh7 cells for end-point infectious titer determination, as described (Milani et al, 2017). The titer value was divided by the titer determined for the LV mixed with medium (the no-serum control) and reported as the percentage of recovery of titer compared to this control.

## Fractionation and sorting of liver cell subpopulations

The mouse liver was perfused (15 mL/min) via the inferior vena cava with 12.5 mL of the following solutions at subsequent steps: (1) PBS EDTA (0.5 mM), (2) HBSS (Hank's balanced salt solution, Gibco) and HEPES (10 mM), (3) HBSS-HEPES 0.03% Collagenase IV (Sigma). The digested mouse liver tissue was harvested, passed through a 100-µm cell strainer (BD Biosciences) and processed into a single-cell suspension. This suspension was subsequently centrifuged three times (30, 25, and 20 × g, for 3 min, at room temperature) to obtain hepatocytes-containing pellets. The nPC-containing supernatant was centrifuged (650 × g, 7 min, at room temperature) and recovered cells were loaded onto a 30/60% Percoll (Sigma) gradient (1800 × g, for 15 min at room temperature). nPC interface was collected and washed twice. The nPC were subsequently incubated with the following monoclonal antibodies: e-fluor 450-conjugated anti-CD45 (30-F11, e-Bioscience), Allophy-cocyanin (APC)-conjugated anti-CD31 (MEC13.3, BD Biosciences), phycoerythrin (PE)-conjugated F4/80 (CI:A3-1, Biorad), PE-Cy5-conjugated anti-CD45R/B220 (from BD Biosciences), PE-Cy7-conjugated anti-CD11c (N418, e-Bioscience). nPC subpopulations (LSEC, KC, pDC) were sorted by FACS, BD FACSAria™ Fusion Cell Sorter (BD Biosciences); the nPC contaminating the PC suspension, were removed by FACS excluding cells labeled by APC-conjugated anti-CD31/anti-CD45 cocktail, thus obtaining sorted hepatocytes (Hep) using BD FACSMelody™ Cell Sorter (BD Biosciences) or MoFlo Astrios EQ Cell Sorter (Beckman Coulter).

## Electron microscopy

Few microliters of concentrated LV batches were absorbed on glow-discharged carbon-coated formvar copper grids and fixed for 20 min with 8% paraformaldehyde in PBS. After several washes in 50 mM glycine in PBS grids were blocked in 1% BSA in PBS and incubated with primary antibodies diluted in blocking buffer for 30–90 min (Anti-CD47 BD Biosciences, 1:10). After several washes in 0.1% BSA in PBS, samples were incubated for 30 min with Protein A-gold (5 or 10 nm), fixed with 1% glutaraldehyde, stained with 2% uranyl acetate, or a mix of 0.4% uranyl acetate and 1.8% methyl-cellulose, then air-dried. Grids were observed with a Zeiss LEO 512 transmission electron microscope. Images were acquired by a 2k × 2k bottom-mounted slow-scan Proscan camera controlled by EsivisionPro 3.2 software. For quantification of labeling density, random images of viral particles were taken at a nominal magnification of 16k, and gold particles associated with virions were manually counted using ImageJ. Virions were defined based on expected size (approximately 120 nm) and an electron-dense core.

## Flow cytometry

Flow cytometry analyses to check human monocyte purity after magnetic sorting were performed using a FACSCanto analyzer (BD Biosciences), equipped with DIVA Software. Between 100,000–500,000 cells were harvested, washed with PBS or MACS buffer (PBS pH 7.2 0.5% bovine serum albumin (BSA), 2 Mm EDTA), treated with fragment crystallizable (Fc) Receptor-Block (Miltenyi Biotec) when antibody stained and then re-suspended in the buffer used for washing. Staining was performed in MACS buffer, incubating cells with antibodies (in the proportion indicated in the table below) for 20 min at 4 °C in the dark. Rainbow beads (BD Biosciences) were used to calibrate the instrument detectors.

 Anti-CD14 PE-Vio770, clone TÜK4 (Miltenyi, 1:20)
 Anti-CD19 PE, clone SJ25C1 (BD Biosciences, 1:50)
 Anti-CD3 APC, clone UCHT1 (BD Biosciences, 1:50)
 Anti-CD16 Pacific Blue, clone 3G8 (BD Biosciences, 1:50).

## Immunohistochemistry imaging

Livers were perfused from the inferior vena cava with PBS EDTA 0.5 mM to remove blood, then harvested and fixed at least 24 h in zinc-formalin. Embedding and cutting of livers and slides preparation, staining, and acquisition were performed by Centro di Imaging Sperimentale facility in San Raffaele Hospital. Briefly, formalin-fixed paraffin-embedded consecutive sections (4 um) were dewaxed and hydrated through graded decrease alcohol series and stained for histology or immunohistochemical characterization (IHC). Slides were immunostained with Automatic Leica BOND RX system (Leica Microsystems GmbH, Wetzlar, Germany). First, tissues were deparaffinized and pre-treated with the Epitope Retrieval Solution (ER1 Citrate Buffer). Anti-GFP primary antibody (Invitrogen, A11122) was incubated 30 min at room temperature (1:1,000) and was developed with Bond Polymer Refine Detection (Leica, DS9800). Slides were acquired with Aperio AT2 digital scanner at a magnification of 20X (Leica Biosystems) and analyzed with Imagescope (Leica Biosystem).

## Immortalized human LSEC characterization

Immortalized human LSEC (Innoprot, 0 https://innoprot.com/product/immortalized-human-hepatic-sinusoidal-endothelial-cells) were characterized by cytofluorimetric analysis of CD31, CD144, CD34, CD45, and CD90 expression and by anti-hFVIII immunofluorescence.

Anti-CD31 APC, clone MEM-05 (Invitrogen, 1:50)

Anti-CD144 PE, clone 16B1 (e-Bioscience, 1:100)

Anti-CD34 PE, clone 4H11 (Immunotools, 1:200)

Anti-CD45 APC, clone HI30 (Invitrogen, 1:100)

Anti-CD90 PE-Cy5, clone 5E10 (BD Pharmigen, 1:200)

Anti-FVIII not conjugated, clone GMA8015 (Green Mountain, 1:100)

Anti-Mouse IgG AF488, polyclonal (Life Technologies, 1:500).

## Imaging flow cytometry acquisition and analysis

One million human primary macrophages or 293T were transduced for 6 h with control fluorescent LV, incubated 1 h at 37 °C with medium, or opsonized fluorescent LV, incubated 1 h at 37 °C with complement-preserved commercial human serum (Sigma) at MOI 10, based on Gag p24 equivalents. This LV dose was selected to avoid direct complement-mediated inactivation of LV. Then, cells were harvested, fixed 20 min using 1% paraformaldehyde solution, stained with hoechst (Invitrogen, 1:2000) for 5 min and then acquired at ImageStream. LV entry in primary human macrophages and in 293 T cells was analyzed by imaging flow cytometry using ImagestreamX MarkII System (Amnis, Cyteck Biosciences). The instrument is equipped with three lasers (405 nm, 488 nm, and 642 nm), a 6-channels CCD camera, Multimag option but no extended depth of field option. Excitation laser settings were the following: 405 nm (10 mW), 488 nm (200 mW). At least 5000 events were collected for each sample with the 60X_0.9NA objective, low speed, and the images were analyzed using IDEAS 6.2 software. Single-stained samples were acquired with the identical laser settings of the samples but without bright-field illumination and side scatter illumination, and were used for compensation. Gating strategy consisted in selecting cells in focus using the Gradient Root Mean Square feature, and then single cells were identified using Area and Aspect Ratio features on the bright-field image. Cells with high Contrast (>25) in the bright-field image (likely apoptotic cells) were excluded from the analysis. In addition, Area of M01 (Hoechst channel) and Aspect Ratio Intensity of M01 features were also used to further eliminate cells with a nucleus not perfectly in focus or with an uneven shape. In order to properly identify the GFP-positive LV signal over background autofluorescence of macrophages in the green channel (Ch02), we created and optimized the LV mask by applying and combining different function masks in IDEAS software. The masking strategy is outlined below. As a first step, a Spot function was used with radius between 0 and 1 and a spot to cell background ratio of 7, then a Peak function was added (spot to cell background ratio value of 1) to refine the selection. In order to eliminate very small and non-specific spots that could be due to high background, a Range function was applied to the Peak mask to keep only objects with area between 2 and 200 pixels and aspect ratio between 0 and 1. An additional Peak function was used, with spot to cell background ratio value of 3.1. The resulting mask was then dilated by 1 pixel and, as a final step, only spots with an area between 0 and 50 pixels and an aspect ratio between 0 and 1 were considered for the analysis (Range function). A mask for the identification of the cytoplasm of the cell was then created and matched with the Spot mask to be sure that we were considering only spots present in the cytoplasmic region of the cell. Finally Spot Count Feature was used

to count the spots. In order to discriminate among active transduction and phagocytosis, the amount of total spots per 100 cells was computed for both macrophages and 293T and then the total obtained from the latter was subtracted from the former. Data were then analyzed in a pair-wise manner (each donor in the two conditions) to account for donor-to-donor variability in basal phagocytosis activity.

## Mice experiments

Founder B6;129S-*F8*tm1Kaz/J mice (referred to as HemoA or *F8* KO) were obtained from The Jackson Laboratories (stock #004424). Founder *F8*tm1Kaz Tg(Alb-F8*R593C)T4Mcal/J mice (referred to as HemoA-R593C; (Bril et al, 2006)) were obtained from The Jackson Laboratories (stock #017706). C57BL/6 and NOD mice were purchased from Charles River Laboratories. All mice were maintained in specific pathogen-free conditions. Vector administration was carried out in male and female adult (7–10-week-old) mice by tail-vein injection (250–500 μL/mouse), while in newborns by temporal vein injection (25–30 μL/mouse). Mice were bled from the retro-orbital plexus using capillary tubes and blood was collected into 0.38% sodium citrate buffer, pH 7.4. Mice were euthanized by $CO_2$ inhalation at the scheduled times. All animal procedures were performed according to protocols approved by the Institutional Animal Care and Use Committee.

## Hemostasis challenge

Mice were anesthetized and the first finger of right foot was cut at the basis. Mice were then returned to their cage and monitored for survival for 24 h.

## Immunofluorescence imaging

Livers were harvested from mice, washed briefly in PBS and fixed 4 h in PBS 4% paraformaldehyde (PFA), then washed again briefly in PBS before being stored at least 24 h in 30% sucrose 0.02% sodium azide in $H_2O$. Livers were the frozen in OCT (optimal cutting temperature) compound (Killik, Bio Optica) and slices 5–20 μm thick were cut at cryostat (Histo-line MC5050), placed on Superfrost® Plus microscope slides (Thermo Scientific) and stored at −80 °C. All the staining steps were performed protected from lights to preserve endogenous fluorescence. Slides were thawed at room temperature for at least 2 h, then washed 3 times 5 min in PBS 0.1% X-Triton. Edges were drawn around the tissue using an immunostaining pap pen (Sigma-Aldrich) to contain staining solutions. Blocking step was performed using PBS 0.1% X-Triton 1% bovine serum albumin (BSA) 5% FBS for 1 h at room temperature in a humid chamber. Blocking solution was then substitute with mix containing primary antibodies in blocking solution and incubation was carried on for 12–16 h at 4 °C in a humid chamber. The primary antibodies mix solution was removed, and slides were washed 3 times 5 min in PBS 0.1% X-Triton. Secondary antibodies were diluted in blocking solution together with Hoechst (Invitrogen, 1:20,000) and incubated for 1 h at room temperature in a humid chamber. Slides were then washed in PBS, and a coverslip was added using Fluoromount-G (Invitrogen) as a mounting medium. Slides were dried for 16 h at

room temperature or for longer at 4 °C, and stored at 4 °C before acquisition. The following antibodies were used for immunofluorescence staining:

Anti-GFP chicken, polyclonal (Abcam, 1:200)

Anti-F4/80 rat, clone CI:A3-1 (Abcam, 1:200)

Anti-Lyve1 rabbit, clone (Novus Biological, 1:250)

Anti-Rabbit IgG Alexa Fluor (AF)647 donkey, polyclonal (Invitrogen, 1:1000)

Anti-Chicken IgY AF488 goat, polyclonal (Invitrogen, 1:1000)

Anti-Rat IgG AF561 goat, polyclonal (Invitrogen, 1:1000).

Images were acquired using confocal microscope Mavig Rs-G4 at ×20 magnification. Images were analyzed using ImageJ or MATLAB software. A custom-written Matlab pipeline was used to analyze the images. Briefly, for all the channels (KC, LSEC, GFP+ and nuclei) we extract black and white masks of the cells/nuclei by background subtraction and then applying a fixed threshold. A mask of the tissue is obtained similarly on the image calculated as max projection across all the channels. Then combining with logical operations the masks of the different cell kinds and the tissue, we are able to estimate the fraction of area occupied by the different cell types as well as the fraction of GFP-positive cells within each population.

## FVIII assays

The concentration of human FVIII was determined in mouse plasma by an enzyme-linked immunosorbent assay (ELISA) specific for human FVIII antigen. Microtiter plates were coated with anti-hFVIII binding Ab (Green Mountain Antibodies #GMA8016, 0.2 μg/well in 0.1 M carbonate buffer, pH 9.6) over night at 4 °C and then blocked 1 h at room temperature with blocking buffer (PBS 0.05% Tween-20, 1 M NaCl, 10% heat-inactivated horse serum, Gibco). Plasma samples are diluted as needed starting from 1:10 in blocking buffer, added to wells (100 μL/well) and incubated 2 h at 37 °C. hFVIII was detected by adding detection Ab (Affinity Biologicals, F8C-EIC-D) 1 h at 37 °C, followed by 5–10 min incubation with 100 μL/well of TMB substrate (Surmodics). Reaction was blocked with HCl 1 N (50 μL/well) and absorbance of each sample was determined spectrophotometrically at 450 nm, using a Multiskan GO microplate reader (Thermo Fisher Scientific) and normalized to antigen standard curve (Moroctocog alfa (ReFACTO or Xyntha), Pfizer, from 25 ng/mL to 0.39 ng/mL serially diluted 1:2 in blocking buffer; dilution was corrected with 10% HemoA murine plasma). hFVIII activity in mouse plasma was measured using Coatest SP FVIII (Chromogenix) following the manufacturer's instructions. Anti-hFVIII Abs were measured in mouse plasma by ELISA. Microtiter plates were coated with Moroctocog alfa (Pfizer, 0.1 μg/well in 0.1 M carbonate buffer, pH 9.6) over night at 4 °C and then blocked 1 h at room temperature with blocking buffer (PBS 0.05% Tween-20, 10% heat-inactivated horse serum, Gibco). Samples are heat-inactivated for 30 min at 56 °C, diluted as needed starting from 1:100 in blocking buffer, added to wells (100 μL/well) and incubated 2 h at 37 °C on orbital shaker. Anti-hFVIII Abs were detected by adding detection Ab (goat anti-mouse IgG-HRP, Sigma, 1:10,000 in blocking buffer) 1 h at 37 °C on orbital shaker, followed by 5–10 min incubation with 100 μL/well of TMB substrate (Surmodics). Reaction was blocked with HCl 1 N (50 μL/well) and absorbance of each sample was determined spectrophotometrically

**The paper explained**

**Problem**

The currently approved gene therapy for the inherited bleeding disease hemophilia A is limited by a decreasing trend in the reconstituted coagulation factor VIII (FVIII) activity and its expected duration is in the range of 5–10 years in adults. In addition, pediatric patients are currently excluded from this type of gene therapy. This is because adeno-associated viral (AAV) vectors do not actively integrate into the genome of target cells and thus are progressively lost upon cell proliferation, such as during liver growth and turnover. Moreover, it remains debated whether ectopic FVIII expression into hepatocytes contributed to the observed outcome.

**Results**

We have engineered integrating lentiviral vectors (LV) to target the FVIII transgene efficiently and specifically to the liver endothelium, by both re-directing LV tropism through LV surface engineering and exploiting the endogenous endothelial-specific FVIII promoter. We report stable long-term FVIII activity in mice with hemophilia A treated as newborns, by a single intravenous administration of the engineered LV targeting liver endothelial cells, the natural site of FVIII production.

**Impact**

In vivo gene therapy with the engineered LV reported here may potentially solve the issues of the currently available gene therapy that is based on AAV vectors targeting FVIII expression to hepatocytes, by providing long-lasting therapeutic benefit after a single administration, even in young individuals.

at 450 nm, using a Multiskan GO microplate reader (Thermo Fisher Scientific) and normalized to standard curve. The standard curve is a pool of 7 different commercial anti-human FVIII Abs raised against different FVIII domains (Green Mountain Antibodies #GMA8002, #GMA8005, #GMA8008, #GMA8011, #GMA8015, #GMA8016, #QED10104) serially diluted 1:2 from 200 ng/mL to 1.56 ng/mL in blocking buffer; dilution was corrected with 1% HemoA murine plasma. Neutralizing anti-human FVIII Abs were determined in heat-inactivated plasma samples (1 h at 56 °C) by Bethesda assay. The test sample was incubated for 2 h at 37 °C with recombinant human FVIII (Moroctocog alfa (ReFACTO or Xyntha), Pfizer, 1 U/mL) with FVIII activity of 100%. Residual FVIII activity was measured using Coatest FVIII SP kit (Chromogenix) and converted into Bethesda Units (BU)/mL, where one BU is defined as the inverse of the dilution factor of the test sample that yields 50% residual FVIII activity.

## FIX assays

The concentration of human FIX was determined in mouse plasma by an enzyme-linked immunosorbent assay (ELISA) specific for human FIX antigen (Asserachrom IX:Ag, Stago) following the manufacturer's instructions. The absorbance of each sample was determined spectrophotometrically, using a Multiskan GO microplate reader (Thermo Fisher Scientific), and normalized to antigen standard curves.

## Statistical analysis

Statistical analyses were performed upon consulting with professional statisticians at the San Raffaele University Center for

Statistics in the Biomedical Sciences (CUSSB). When normality assumptions were not met, non-parametric statistical tests were performed. The two-tailed Mann–Whitney test was performed to compare two independent groups, while in the presence of more than two independent groups, the Kruskal–Wallis test was followed by post hoc analysis (Dunn's test for multiple comparisons against the reference control group along with Bonferroni's correction) was applied. For repeated measures over time, two-way ANOVA was performed. For paired observations, the Wilcoxon matched-pairs test was performed. Inferential techniques were applied in the presence of adequate sample sizes ($n \geq 5$), otherwise only descriptive statistics are reported.

## For more information

Author's websites: https://research.hsr.it/en/institutes/san-raffaele-telethon-institute-for-gene-therapy.html; https://www.telethon.it; Coagulation factor VIII: https://www.omim.org/entry/300841?search=%22factor%20viii%22&highlight=%22factor%20viii%22; Hemophilia A: https://www.orpha.net/en/disease/detail/98878?name=hemophilia%20A&mode=name.

## Data availability

Raw images shown in Fig. 3D can be accessed in BioImage Archive with the following accession code: S-BIAD1121.

The source data of this paper are collected in the following database record: biostudies:S-SCDT-10_1038-S44321-024-00072-8.

## Peer review information

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

## Acknowledgements

This work was supported by Fondazione Telethon SR-Tiget Core Grant TTACC0422TT to AC, EU Horizon 2020 Program (825825 UPGRADE) to LN, Bioverativ/Sanofi sponsored research agreement to LN, GeneSpire sponsored research agreement to AC. The authors thank CUSSB (Vita-Salute San Raffaele University) for statistical consulting, Eugenia Cammarota and the Advanced Light and Electron Microscopy Bioimaging Center (ALEMBIC, San Raffaele Scientific Institute), Amleto Fiocchi and the Experimental Imaging Center (San Raffaele Scientific Institute) for imaging techniques and analysis. The authors thank Sofia Ottonello for technical help and all the members of the Cantore laboratory for helpful discussions. CC conducted this study as a partial fulfillment of his International PhD Course in Molecular Medicine at San Raffaele University, Milan.

## Author contributions

**Michela Milani**: Conceptualization; Data curation; Formal analysis; Supervision; Investigation; Visualization; Methodology; Writing—original draft. **Cesare Canepari**: Data curation; Formal analysis; Investigation; Methodology; Writing—review and editing. **Simone Assanelli**: Formal analysis; Investigation; Visualization; Methodology; Writing—review and editing. **Simone Merlin**: Supervision; Investigation; Methodology. **Ester Borroni**: Formal analysis; Investigation; Methodology; Writing—review and editing. **Francesco Starinieri**: Formal analysis; Investigation; Visualization; Methodology; Writing—review and editing. **Mauro Biffi**: Formal analysis; Investigation; Methodology; Writing—original draft. **Fabio Russo**: Investigation; Methodology. **Anna Fabiano**: Formal analysis; Investigation; Methodology. **Désiree Zambroni**: Formal analysis; Investigation; Methodology; Writing—review and editing. **Andrea Annoni**: Supervision; Investigation; Methodology. **Luigi Naldini**: Supervision; Funding acquisition; Project administration; Writing—review and editing. **Antonia Follenzi**: Resources; Supervision; Methodology; Project administration; Writing—review and editing. **Alessio Cantore**: Conceptualization; Supervision; Funding acquisition; Writing—original draft; Project administration.

Source data underlying figure panels in this paper may have individual authorship assigned. Where available, figure panel/source data authorship is listed in the following database record: biostudies:S-SCDT-10_1038-S44321-024-00072-8.

## Disclosure and competing interests statement

LN, AC, AA, MM, AFo, and SM are inventors on patent applications submitted by Fondazione Telethon, San Raffaele Scientific Institute, Bioverativ/Sanofi or University of Piemonte Orientale on LV technology related to the work presented in this manuscript. Fondazione Telethon and San Raffaele Scientific Institute through SR-Tiget, have established a research collaboration on liver-directed lentiviral gene therapy of hemophilia with GeneSpire. Luigi Naldini is a member of the journal's editorial board. This has no bearing on the editorial consideration of this article for publication. The remaining authors declare no competing interests.

# Expanded View Figures

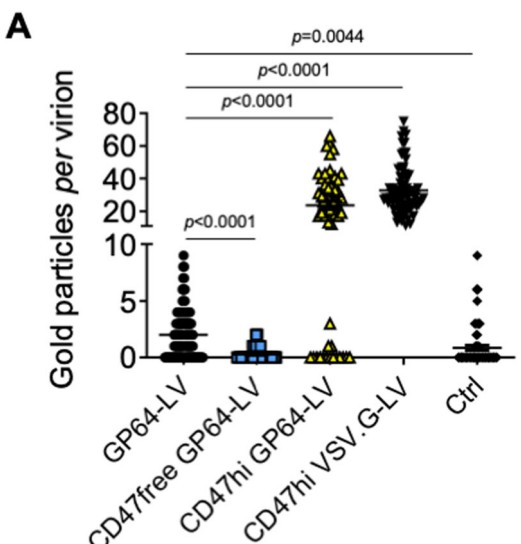

**Figure EV1.  Electron microscopy analysis of CD47 content on LV particles.**

(A) Single values, mean and SEM of gold particles per virion of LV batches produced by control (GP64-LV), CD47-overexpressing (CD47hi GP64-LV or VSV.G-LV), or CD47-negative 293T cells (CD47free GP64-LV), immunostained with anti-CD47 antibody, or as staining control without the primary antibody (Ctrl) and analyzed by electron microscopy ($n = 57$–114 virions per sample). Kruskal–Wallis test with Dunn's multiple comparison test (compared to GP64-LV).

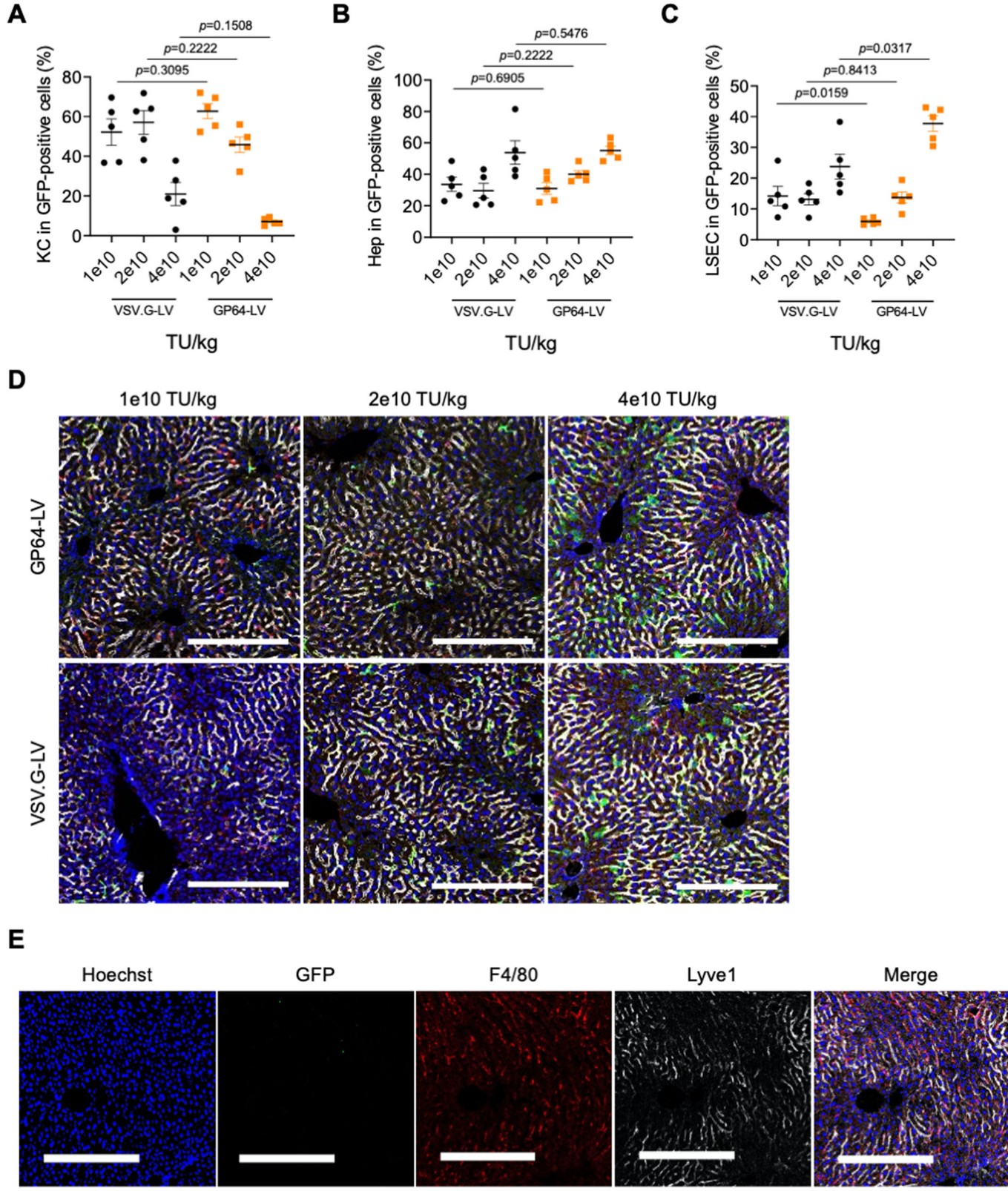

◀ **Figure EV2. Evaluation of GP64-LV liver transduction in vivo in mice.**

(**A–C**) Single values, mean and SEM of the % of KC (**A**), hepatocytes (**B**) or LSEC (**C**) in GFP-positive cells shown in Fig. 3C, reported here separately for statistical analysis. $n = 5$ per group. Mann–Whitney test. (**D**). Representative merge images of immunofluorescent staining of liver sections analyzed in Fig. 3B, C at the indicated doses. Red: F4/80-positive cells (KC); White: Lyve1-positive cells (LSEC); Green: GFP (transduced cells); Blue: Hoechst (nuclei). Scale bar: 250 μm. Images of the 4e10 TU/kg LV dose are also presented in Fig. 3D. (**E**) Representative images of immunofluorescent staining of a liver section from an untreated control mouse. Red: F4/80-positive cells (KC); White: Lyve1-positive cells (LSEC); Green: GFP (transduced cells); Blue: Hoechst (nuclei). Scale bar: 250 μm.

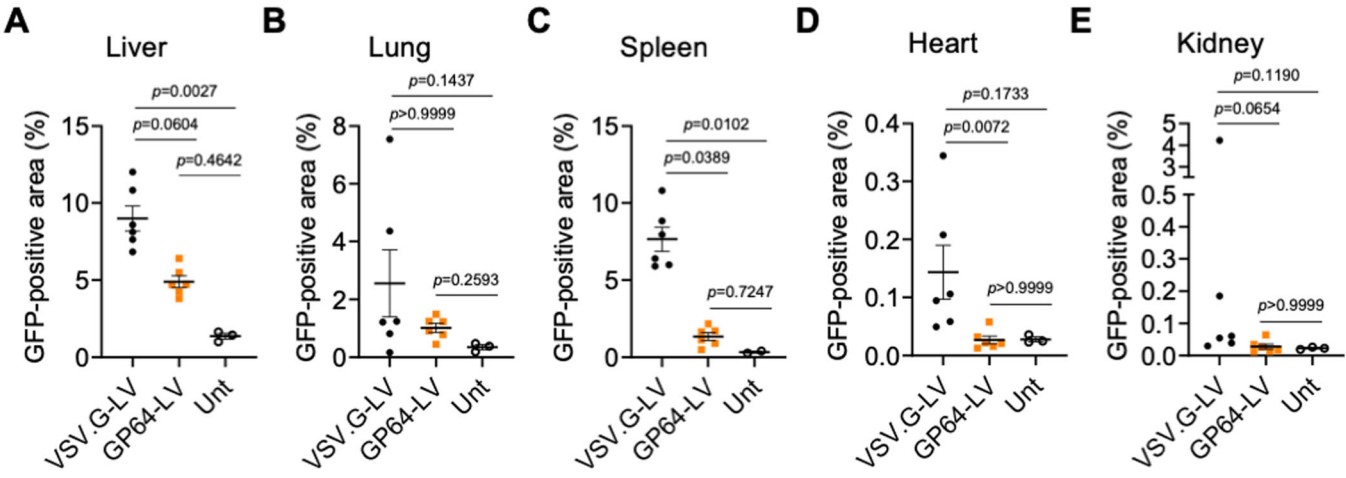

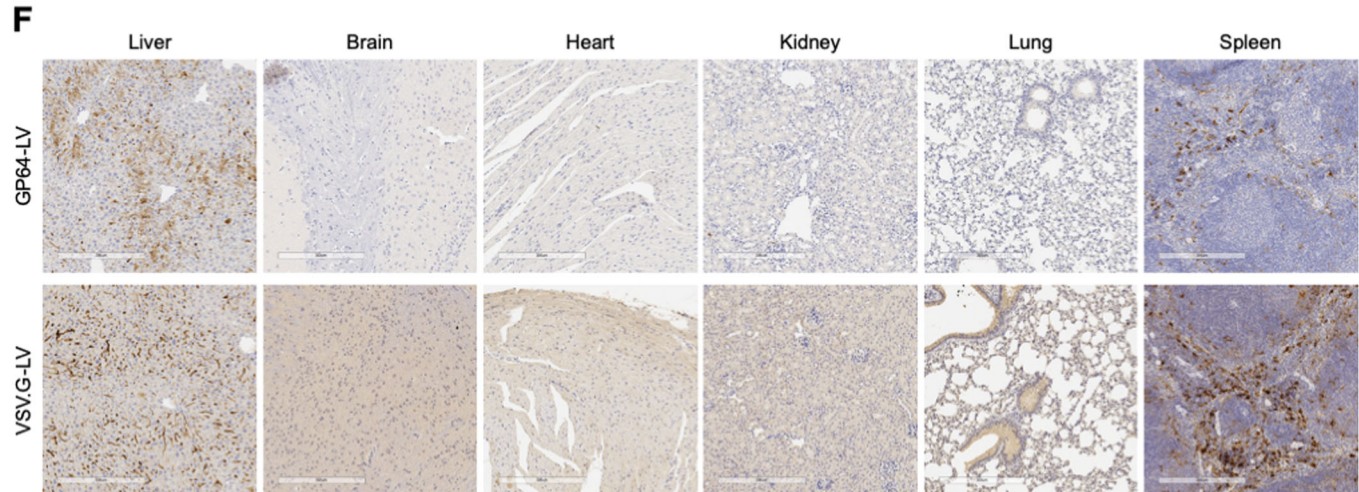

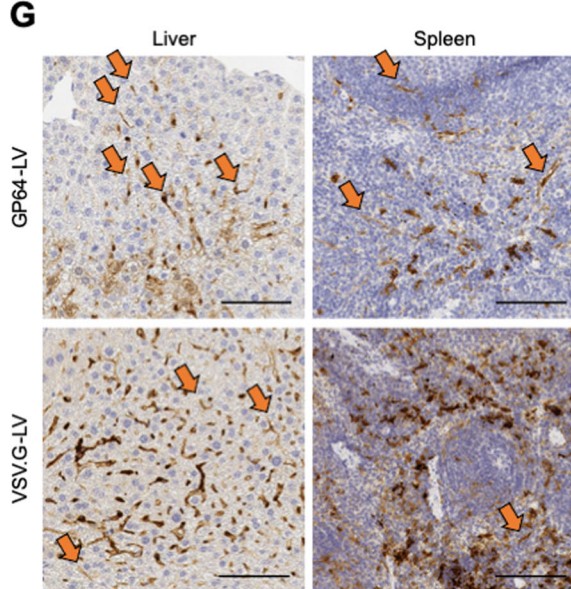

◀ **Figure EV3. Biodistribution analysis of GP64-LV after systemic administration in mice.**

(**A–E**) Single values, mean and SEM of GFP-positive tissue measured by immunohistochemistry in the liver (**A**), lung (**B**), spleen (**C**), heart (**D**) or kidney (**E**) of 8-week-old mice analyzed 10 days after i.v. administration of VSV.G-LV or GP64-LV at 4e10 TU/kg ($n = 6$ mice per group), or left untreated for background signal evaluation ($n = 3$). Kruskal–Wallis test with Dunn's multiple comparison test. (**F**) Representative anti-GFP immunohistochemistry images of liver, brain, heart, kidney, lungs, spleen of mice treated with GP64-LV or VSV.G-LV, as indicated, analyzed in (**A–E**). Scale bar: 300 µm. (**G**) Higher magnification of liver and spleen sections shown in (**F**). Orange arrows indicate GFP-positive endothelial cells, identified by morphology. Scale bar: 100 µm.

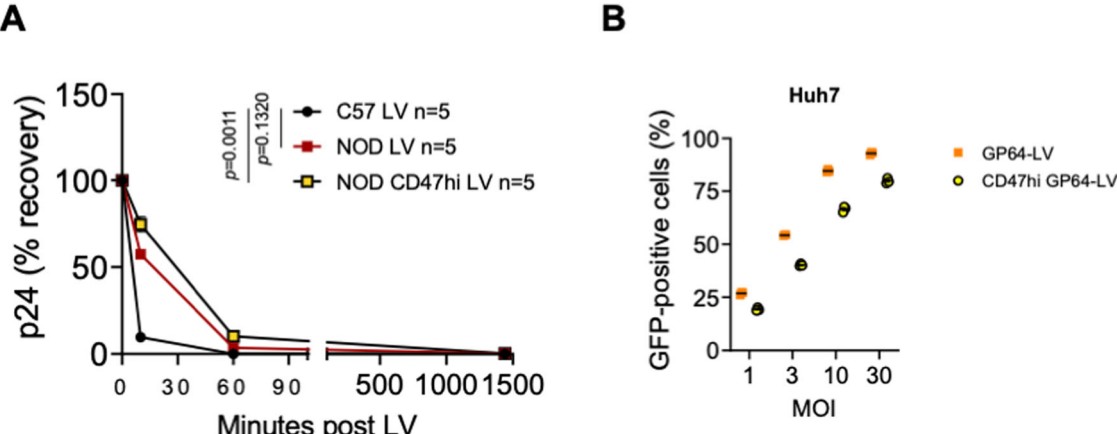

**Figure EV4.  GP64-LV half-life after systemic administration in mice.**

(A) Percentage of the serum concentrations of LV particles (measured as HIV Gag p24) recovered at the indicated time (minutes) after administration of GP64-LV or CD47hi GP64-LV relative to the total amount of administered LV particles to C57BL/6 or NOD mice, as indicated (8e10 TU/kg). Kruskal–Wallis test with Dunn's multiple comparison test (compared to C57 LV). (B) Single values, mean, and SEM of the percentage of GFP-positive Huh7 cells, transduced with GP64-LV or CD47hi GP64-LV, at the indicated MOI ($n = 3$).

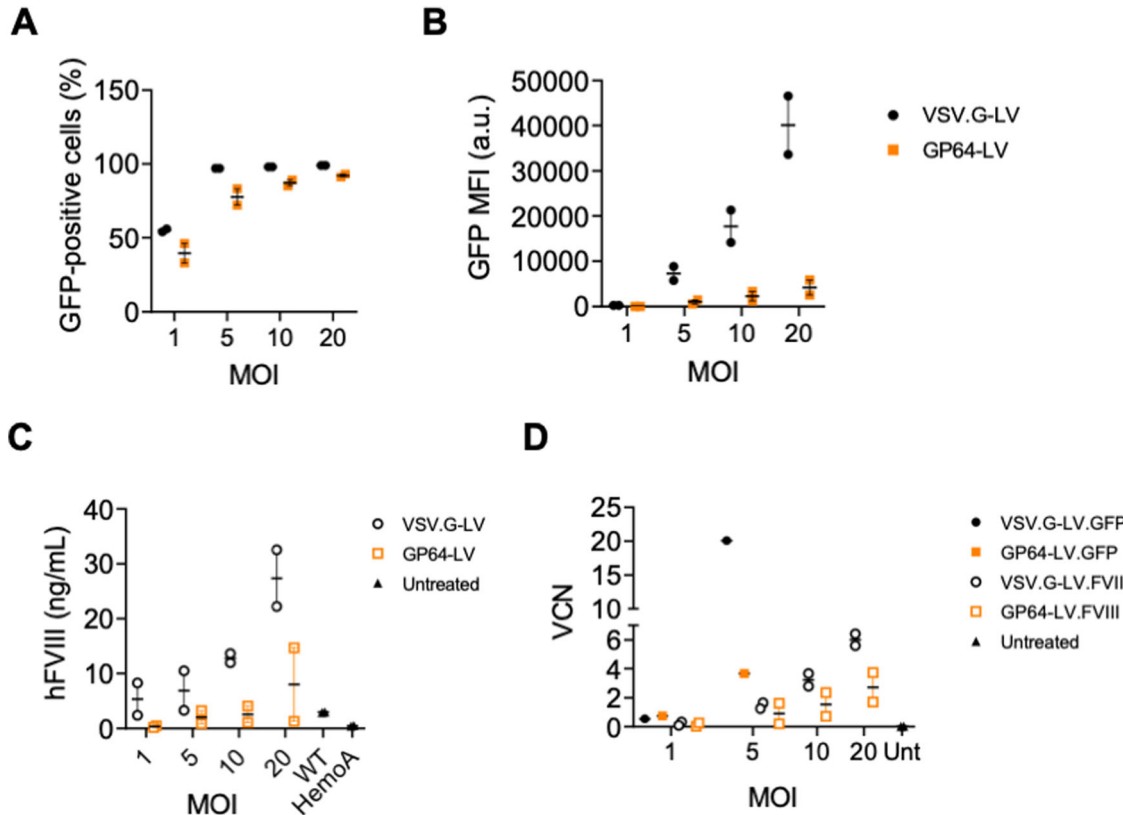

**Figure EV5. Ex vivo transduction of primary human blood outgrown endothelial cells.**

(A, B) Mean and SEM with single values of the % (C) or MFI (D) of GFP-positive primary blood outgrowth endothelial cells (BOEC) derived from a hemophilia A donor (*n* = 2 independent experiments) transduced with VSV.G-LV or GP64-LV at the indicated MOI. (C) Mean and SEM with single values of the amount of hFVIII measured in the culture supernatant of primary BOEC derived from a hemophilia A donor (*n* = 2 independent experiments) transduced with VSV.G-LV or GP64-LV at the indicated MOI, or left untreated. hFVIII concentration reported is obtained from two supernatant collection. (D) Mean and SEM with single values of the VCN of reversed transcribed LV genome of the primary BOEC (shown in (A–C)) transduced with VSV.G-LV or GP64-LV encoding for GFP or hFVIII at the indicated MOI, or left untreated.

