## [Peer Review File · EMBO Molecular Medicine]

GP64-pseudotyped lentiviral vectors target liver endothelial cells and correct hemophilia A mice

Alessio Cantore, Michela Milani, Cesare Canepari, Simone Assanelli, Simone Merlin, Ester Borroni, Francesco Starinieri, Mauro Biffi, Fabio Russo, Anna Fabiano, Désiree Zambroni, Andrea Annoni, Luigi Naldini, and Antonia Follenzi

Corresponding author: Alessio Cantore (cantore.alessio@hsr.it)

Review Timeline:

Submission Date:	6th Sep 23
Editorial Decision:	28th Sep 23
Revision Received:	21st Mar 24
Editorial Decision:	4th Apr 24
Revision Received:	15th Apr 24
Accepted:	16th Apr 24

Editor: Zeljko Durdevic

Transaction Report:

28th Sep 2023

Dear Dr. Cantore,

Thank you for the submission of your manuscript to EMBO Molecular Medicine. We have now received feedback from the two reviewers who agreed to evaluate your manuscript. As you will see from the reports below, the referees acknowledge the interest of the study but also raise serious concerns particularly regarding, but not limited to, the inconsistencies of the data presented and the weak in vivo effect.

Taking the referee reports in consideration, it is clear that publication of the paper cannot be considered at this stage. I note that addressing the reviewers concerns in full appears to require a lot of additional work and experimentation. I am unsure whether you will be able or willing to address those and return a revised manuscript within the 6 months deadline. If you would like to discuss further the points raised by the referees, I am available to do so via email or video. Let me know if you are interested in this option.

Please note that further consideration of a revision that addresses reviewers' concerns in full will entail a second round of review. EMBO Molecular Medicine encourages a single round of revision only and therefore, acceptance or rejection of the manuscript will depend on the completeness of your responses included in the next, final version of the manuscript. For this reason, and to save you from any frustrations in the end, I would strongly advise against returning an incomplete revision and would also understand your decision if you chose to rather seek rapid publication elsewhere at this stage.

Should you find that the requested revisions are not feasible within the constraints outlined here and choose, therefore, to submit your paper elsewhere, we would welcome a message to this effect.

Yours sincerely,

Zeljko Durdevic

We require:

- 1) A .docx formatted version of the manuscript text (including legends for main figures, EV figures and tables). Please make sure that the changes are highlighted to be clearly visible.
- 2) Individual production quality figure files as .eps, .tif, .jpg (one file per figure). For guidance, download the 'Figure Guide PDF' (<https://www.embopress.org/page/journal/17574684/authorguide#figureformat>).
- 3) A .docx formatted letter INCLUDING the reviewers' reports and your detailed point-by-point responses to their comments. As part of the EMBO Press transparent editorial process, the point-by-point response is part of the Review Process File (RPF), which will be published alongside your paper.
- 4) A complete author checklist, which you can download from our author guidelines (<https://www.embopress.org/page/journal/17574684/authorguide#submissionofrevisions>). Please insert information in the checklist that is also reflected in the manuscript. The completed author checklist will also be part of the RPF.

6) It is mandatory to include a 'Data Availability' section after the Materials and Methods. Before submitting your revision, primary datasets produced in this study need to be deposited in an appropriate public database, and the accession numbers and database listed under 'Data Availability'. Please remember to provide a reviewer password if the datasets are not yet public (see <https://www.embopress.org/page/journal/17574684/authorguide#dataavailability>).

7) For data quantification: please specify the name of the statistical test used to generate error bars and P values, the number (n) of independent experiments (specify technical or biological replicates) underlying each data point and the test used to calculate p-values in each figure legend. The figure legends should contain a basic description of n, P and the test applied. Graphs must include a description of the bars and the error bars (s.d., s.e.m.).

8) We would also encourage you to include the source data for figure panels that show essential data. Numerical data should be provided as individual .xls or .csv files (including a tab describing the data). For blots or microscopy, uncropped images should be submitted (using a zip archive if multiple images need to be supplied for one panel). Additional information on source data and instruction on how to label the files are available at

13) Author contributions: the contribution of every author must be detailed in a separate section (before the acknowledgments).

14) A Conflict of Interest statement should be provided in the main text.

15) Every published paper now includes a 'Synopsis' to further enhance discoverability. Synopses are displayed on the journal webpage and are freely accessible to all readers. They include a short stand first (maximum of 300 characters, including space) as well as 2-5 one-sentence bullet points that summarize the paper. Please write the bullet points to summarize the key NEW findings. They should be designed to be complementary to the abstract - i.e. not repeat the same text. We encourage inclusion of key acronyms and quantitative information (maximum of 30 words / bullet point). Please use the passive voice. Please attach these in a separate file or send them by email, we will incorporate them accordingly.

***** Reviewer's comments *****

Referee #1 (Remarks for Author):

The aim of this study was to evaluate properties of GP64 pseudotyped lentivectors (GP64-LV) and compare these properties to the more standard VSV.G-LV envelope.

Initial studies were performed to standardize LV doses and to assess resistance to complement inactivation. These results confirm a significant degree of protection against complement inactivation mediated by the GP64-LV envelope.

Subsequent studies were focused on the cellular tropism of GP64-LV and VSV.G-LV. A range of in vitro and in vivo assessments were performed that showed with varying levels of clarity that GP64-LV demonstrated increased tendencies for transduction of LSECs as opposed to hepatocytes.

Overall, this manuscript describes a comprehensive evaluation of the relative complement sensitivity and cellular tropisms of these two LV envelopes. Some new and biologically important information is presented in this manuscript, but certain aspects of the study outcomes require significant clarification and re-stating.

Editorial review and suggestions are as follows (not in order of priority) -

a) The authors have noted that their efforts to inhibit LV phagocytosis by inserting CD47 into the particle envelope are proving effective, but they also report that CD47 expression on the GP64 envelope is heterogeneous and absent in some fraction of the vector particles. Do the authors have data that evaluates whether transduction efficacy of GP64-LV was influenced by CD47 expression?

b) The evidence that GP64-LV has a higher tropism for LSECs than VSV.G-LV doesn't appear consistent. In Fig 3C (GFP+ve tissue) the rates of LSEC +ve results is not very different at any of the vector titers between GP64-LV and VSV.g-LV. Fig 3D doesn't provide further clarification concerning the relative hepatocyte vs Kupffer cell vs LSEC tropism. Finally, fig 3 F (that should be labelled as showing on only GP64-LV data) indicates marked differences in VCN with GP64 between hepatocytes (very low) and LSECs (markedly increased) - this is in marked contrast to the data in Fig 3C. This is confusing and requires more interpretive discussion.

c) The results of FIX transgene expression don't provide much additional benefit in the overall theme of the report. They are almost a distraction from the main message of enhanced delivery to LSECs. I would suggest either providing this in a distinct section of the report or even possibly moving it to a supplement. This confusion is further compounded by the data in Fig 4 where good FIX expression is documented in the face of very low hepatocyte VCNs. More explanation of this outcome is required. Lastly, the data in Fig 5 A-D show that although earlier results (eg. Fig 3C) show similar positivity for hepatocyte expression with both vectors there is a marked benefit with VSV.G-LV in these primary hepatocyte transductions. Again, these findings appear to be inconsistent and require more commentary.

d) Fig 5 E-H show more data that seems confusing. In these studies with immortalized LSECs there is very little difference in any of the measured parameters between the two vectors. GFP +ve cells, MFI, FVIII expression and VCNs all seem relatively similar for GP64 and VSV.G, thus arguing against a significant tropism advantage of GP64 for LSEC delivery. These results

need clarification if the tropism theme is to be supported.

e) Lastly in Fig 6A there appears to be no FVIII expression in any of the VSV.G-LV mice despite the fact that results elsewhere in the report (Fig 3C, 5G etc) show LSEC transduction with this vector and in 5G some level of FVIII production by LSECs in vitro.

f) Also, in Fig 6A, B and C there needs to be more explanation of the results with GP64. Two mice appear to have much higher and persistent FVIII expression than the other 3 mice - is this related to anti-FVIII Ab development - it's not clear from Fig 6C. Also, why did any of these mice develop a FVIII immune response as prior studies with these transgenic animals have demonstrated strong immune tolerance to FVIII that is very difficult to break. How did that happen here?

g) Why are the levels of FVIII in the neonates so low and again there appears to be some evidence of a late break in tolerance in some animals.

h) The LSECs studied in culture were obtained as immortalized cells (how were they immortalized?). The ex vivo phenotype of these cells is notoriously very fragile - how did the authors confirm a consistent LSEC phenotype in these studies?

Overall, this reviewer needs to see more commentary/explanation of the inconsistencies reported in this manuscript. Currently the study conclusions re. enhanced LSEC tropism with GP64 are not adequately supported - this may be, in part, due to the presentation of the evidence.

Referee #2 (Remarks for Author):

In this manuscript, Milani and colleagues reported that using GP64-pseudotyped lentivirus (GP64-LV) transduced liver sinusoidal endothelial cells (LSECs) better than VSVG-LV and combining phagocytosis inhibitor CD47 could further improve the transduction efficiency of GP64-LV on LSECs. using GP64-LV to deliver FVIII transgene to LSECs could achieve stable therapeutic levels of FVIII and correct the bleeding phenotype of hemophilia A mice. There are concerns in this manuscript that need to be addressed.

Major concerns

- The title of this manuscript emphasizes the efficacy of GP64-LV transducing LSECs in hemophilia A gene therapy, but the data from hemophilia A model studies were very light and weak.
- In Figure 2, Figure 2B: were there statistically significant differences between those groups? The N was low. Were those three data points from three patients? Could they include all data points from 6 patients and increase N in 293T experiments? Figure 2C shows the fresh serum opsonized phagocytosis efficiency of LV in macrophages. This result is contradictory to their previous report (Milani et al. 2017), which showed that complement components in fresh serum could inactivate LV.
- Figure 3D: The authors should provide untransduced liver tissue as a control for staining in parallel. Representative images represent how many biological repeats?
- There was a discrepancy between the data shown in Figure 3A versus the text on page 7. There was no difference in VCN between the 2×10^{10} and 4×10^{10} GP64-LV groups, but it was described as an LV-dose-dependent increase.
- It may not be appropriate to use the VSVG-LV FIX data from their previous report, Milani et al. 2019, to compare the results from the current GP64-LV FIX study (figure 3E). It would be more rigorous if they conducted VSVG-LV and GP64-LV FIX studies in parallel.
- Fig EV3F: it is difficult to tell whether endothelial cells were transduced. It was unclear what was stained in those sections. The authors should provide some information for readers to understand what was stained in their immunohistochemistry images. Also, providing some zoom-in areas with higher magnification images may help.
- Figure 5: There were only 2 data points in many conditions, and some have huge variations between the two data points. The authors should increase the N number to {greater than or equal to}3 before drawing a conclusion.
- The authors stated that FVIII expression was detected in hemophilia A mice treated with GP64-LV-F8.coFVIII, but undetectable in animals injected with 5×10^{10} TU/kg of VSVG-LV-F8co.FVIII. However, their previous study (Merlin et al. Mol Ther 2017) using VSVG-LV-F8 showed that all animals injected with 10^9 TU VSVG-LV-F8 had sustained FVIII expression. The author should commend the discrepancy in the results from their current vs. previous studies. If it was because of the quality of viruses made at different times, they should not use previous data from the VSVG-LV-FIX study to compare their current GP64-LV-FIX studies (Figure 3E).
- The authors stated that anti-FVIII antibodies did not neutralize FVIII activity (page 10, lines 6-7), but they only showed anti-FVIII total IgG (Figure 6C), which cannot distinguish the inhibitory vs. not-inhibitory antibodies. They did not provide inhibitor titers. Did the VSVG-LV group develop inhibitory antibodies?
- In the neonatal model study, there was no significant difference in the survival rate in the hemostasis challenge between the GP64-LV group and the untreated or VSVG group. How long after the treatment the hemostasis challenge was performed? The authors may want to increase the number of mice or increase the dose of the virus to treat animals to strengthen their data to support their conclusion that GP64-LV works better than VSVG-LV.

Minor concerns

- In Figure 1, Figure 1A: the legend for plot data was missing. Which one depicts VSVG-LV, and which one is GP64-LV? Figure 1D, the Y-axis title does not make sense. It was unclear Fold of what? Figure E-I: it was unclear how long the incubation time was.
- Fig EV1: gold particles per virion in the GP64-LV, CD47free GP64-LV, and control groups all were very low. Could the authors make a separate figure or two segments in Y-axis to zoom in on those low numbers so readers can get a better view of those data points?
- Fig EV2D: the authors should provide panels from each fluorescent staining and then the merged image, as they did in Figure 3D, so that readers can see more detailed information about the transduction efficiency in each cell type.
- Fig EV4: it would be helpful to show the standard deviation in each group.
- Page 22, "ReFACTO" should be changed to "Xyntha".

We want to thank the Reviewers for their positive appraisal of our manuscript and constructive comments. In order to address them and further support our conclusions we have performed additional experimental work, which has been included in the revised figures and text, and modified the manuscript according to the Reviewers' suggestions. All the changes have been highlighted in red in the revised manuscript. We think that the manuscript has been improved by the revisions and for that we are grateful.

Please find below a detailed point-by-point response to each Reviewer's remark.

Referee #1 (Remarks for Author):

The aim of this study was to evaluate properties of GP64 pseudotyped lentivectors (GP64-LV) and compare these properties to the more standard VSV.G-LV envelope.

Initial studies were performed to standardize LV doses and to assess resistance to complement inactivation. These results confirm a significant degree of protection against complement inactivation mediated by the GP64-LV envelope.

Subsequent studies were focused on the cellular tropism of GP64-LV and VSV.G-LV. A range of *in vitro* and *in vivo* assessments were performed that showed with varying levels of clarity that GP64-LV demonstrated increased tendencies for transduction of LSECs as opposed to hepatocytes.

Overall, this manuscript describes a comprehensive evaluation of the relative complement sensitivity and cellular tropisms of these two LV envelopes. Some new and biologically important information is presented in this manuscript, but certain aspects of the study outcomes require significant clarification and re-stating.

Editorial review and suggestions are as follows (not in order of priority) –

a) The authors have noted that their efforts to inhibit LV phagocytosis by inserting CD47 into the particle envelope are proving effective, but they also report that CD47 expression on the GP64 envelope is heterogeneous and absent in some fraction of the vector particles. Do the authors have data that evaluates whether transduction efficacy of GP64-LV was influenced by CD47 expression?

We thank the Reviewer for raising this question. To answer the Reviewer's question, we transduced a hepatocyte cell line (Huh7) with GP64-LV or CD47-hi GP64-LV at 4 different matched doses, in triplicate. We observed a slightly lower percentage of GFP-positive cells when using the latter vector compared to the former. These data suggest that the increased transduction of LSEC *in vivo* was not due to a generally higher transduction efficiency of CD47hi GP64-LV compared to control GP64-LV, but rather to higher protection from uptake by professional phagocytes *in vivo* and subsequent higher availability for transduction of LSEC. We have now included the new data in Expanded View Figure 4 and described them in the Results section of the revised manuscript.

b) The evidence that GP64-LV has a higher tropism for LSECs than VSV.G-LV doesn't appear consistent. In Fig 3C (GFP+ve tissue) the rates of LSEC +ve results is not very different at any of the vector titers between GP64-LV and VSV.g-LV.

The statistical analysis of Fig 3C is reported in Expanded View Fig 2A-C. In particular, in panel C, we performed a statistical analysis on the percentage of LSEC (Lyve1-positive) among GFP-positive cells in the liver of GP64-LV-treated mice compared to VSV.G-LV-treated mice at 3

different doses. The results indicate a significantly higher percentage of GFP-positive LSEC in GP64-treated compared to VSV.G-treated mice at 4e10 TU/kg. We have now added a sentence to highlight this point in the Results section of the revised manuscript.

Fig 3D doesn't provide further clarification concerning the relative hepatocyte vs Kupffer cell vs LSEC tropism.

We thank the Reviewer for this comment. We have now included higher magnifications of the images and highlighted GFP-positive LSEC and Kupffer cells in Fig 3D of the revised manuscript.

Finally, fig 3 F (that should be labelled as showing on only GP64-LV data) indicates marked differences in VCN with GP64 between hepatocytes (very low) and LSECs (markedly increased) - this is in marked contrast to the data in Fig 3C. This is confusing and requires more interpretive discussion.

We thank the Reviewer for this comment, as it helps improve the clarity of the data presentation. We have now expanded and rephrased the description of the former Fig 3F (now Fig 4B) in the Results section of the revised manuscript. Please note that VCN measurement provides better resolution compared to GFP imaging in the analysis of transduction of different liver cell types. Indeed, a similar GFP-positive area may correspond to cells carrying different numbers of LV copies, thus the data in former Fig 3F, now Fig 4B, suggest that the LSEC transduced by GP64-LV have multiple LV copies *per cell*, while transduced hepatocytes likely carry a single LV copy. This is particularly relevant for secreted transgenes such as clotting factors, since a cell transduced at multiple LV copies results in higher transgene output in the bloodstream compared to a cell transduced with a single LV copy. Indeed, the high VCN in LSEC achieved by GP64-LV prompted us to further evaluate GP64-LV in the context of FVIII production. As requested by the Reviewer, we provide additional discussion related to this point in the revised manuscript.

c) The results of FIX transgene expression don't provide much additional benefit in the overall theme of the report. They are almost a distraction from the main message of enhanced delivery to LSECs. I would suggest either providing this in a distinct section of the report or even possibly moving it to a supplement.

We thank the Reviewer for this suggestion. We have now moved panels E and F of Figure 3 into revised Figure 4, to separate data with GFP as transgene (Figure 3) from data with FIX as transgene (Figure 4) or with FVIII as transgene (Figure 6).

This confusion is further compounded by the data in Fig 4 where good FIX expression is documented in the face of very low hepatocyte VCNs. More explanation of this outcome is required.

We would like to point out that low VCN in purified hepatocytes reflects the fact that a fraction of all hepatocytes is transduced, and that likely most transduced hepatocytes carry a single LV copy. However, hepatocytes represent around 75% of liver cells, thus, even if LV copies *per cell* are low, the overall amount of transduction allows for FIX output, up to 20% of normal output, at the highest LV dose tested. In addition, FIX expression is highly optimized, and we expect high production *per cell*, thanks to the strength of the promoter, combined with codon optimization of the FIX transgene sequence. These data are in line with our previous reports including data on hepatocyte VCN and FIX output in mice treated with VSV.G-LV (Milani *et al.* Sci Transl Med 2019). As requested by the Reviewer, we have now included a sentence in the Results section of the revised manuscript, to better clarify this point.

Lastly, the data in Fig 5 A-D show that although earlier results (eg. Fig 3C) show similar positivity for hepatocyte expression with both vectors there is a marked benefit with VSV.G-LV in these primary hepatocyte transductions. Again, these findings appear to be inconsistent and require more commentary.

We would like to point out that the data reported in Fig 5A-D are in line with the dose-response reported in former Fig 3E, now Fig 4A, showing much higher hepatocyte-derived FIX output in VSV.G-LV-treated mice compared to GP64-LV-treated mice, indicating higher transduction of mouse hepatocytes *in vivo*, by the former LV. Please note that Fig 3C shows how the percentage of GFP-positive area is distributed among the main liver cell types and as mentioned above, the percentage of GFP-positive area does not inform about LV copies *per cell* (i.e., VCN), which however greatly impact FIX or FVIII output. Thus, taken together, the data presented suggest that at increasing LV doses, VSV.G-LV more efficiently transduce mouse hepatocytes than GP64-LV *in vivo*, as well as *in vitro*, while GP64-LV more efficiently transduce LSEC compared to hepatocytes, *in vivo*, as well as *in vitro*. Moreover, please consider that the data shown in Fig 5 A-D are obtained from primary human cells, thus suggesting that the difference between the two LV pseudotypes is even more marked in human cells. As requested by the Reviewer, we provide additional discussion related to this point in the revised manuscript.

d) Fig 5 E-H show more data that seems confusing. In these studies with immortalized LSECs there is very little difference in any of the measured parameters between the two vectors. GFP+ve cells, MFI, FVIII expression and VCNs all seem relatively similar for GP64 and VSV.G, thus arguing against a significant tropism advantage of GP64 for LSEC delivery. These results need clarification if the tropism theme is to be supported.

We thank the Reviewer for raising the point. We have performed additional experiments with human primary LSEC and have now included the data in Fig 5 E-H of the revised manuscript. The data overall confirm similar to slightly higher transduction of LSEC by GP64-LV compared to VSV.G-LV. We think that the interpretation of these results should be combined with the lower hepatocyte transduction by GP64-LV compared to VSV.G-LV, shown in Fig 5A-D. Our interpretation is the following: *in vivo* administered LV circulate through liver sinusoids, then i) VSV.G-LV can efficiently transduce both hepatocytes and LSEC, resulting in high hepatocyte transduction; ii) on the other hand, since GP64-LV transduce hepatocytes at lower efficiency than VSV.G-LV, most of them enter and transduce LSEC, *in vivo*. This interpretation is consistent with both *in vitro* and *in vivo* data, showing lower efficiency of gene transfer into hepatocytes by GP64-LV compared to VSV.G-LV, paralleled by similar (*in vitro*) or even higher (*in vivo*) gene transfer into LSEC. Furthermore, please note that *in vitro* we only have a single cell type at the time, showing transduction efficiency at matched doses of the two LV, while *in vivo* there are multiple cell types in the liver available for transduction at the same time. We have now revised the text to better reflect the available results in the revised manuscript.

e) Lastly in Fig 6A there appears to be no FVIII expression in any of the VSV.G-LV mice despite the fact that results elsewhere in the report (Fig 3C, 5G etc) show LSEC transduction with this vector and in 5G some level of FVIII production by LSECs *in vitro*.

We would like to point out that transduction efficiency also depends on vector infectivity and transgene output also depends on promoter strength. In the experiment reported in Fig 3C, we use a PGK.GFP LV, which titers very well and is highly infectious. By contrast, in the experiment reported in Fig 6A, we used FVIII-carrying LV that are less infectious than GFP-expressing LV, and we used the endogenous F8 promoter, which is a weak promoter. In Fig 5G, we used the same

vector and construct used for the *in vivo* experiments reported in Fig 6, and indeed FVIII production is very low, as for GP64-LV. Thus, since GP64-LV transduce LSEC at higher VCN than VSV.G-LV (see Fig 6F), it is likely that the FVIII output from LSEC transduced by VSV.G-LV at low VCN is below detection in this experiment. These data are in accordance with data obtained in a new experiment performed in newborn hemophilia A mice at higher LV doses (see below), showing well-detectable but lower circulating FVIII in VSV.G-LV-treated mice compared to GP64-LV-treated mice.

f) Also, in Fig 6A, B and C there needs to be more explanation of the results with GP64. Two mice appear to have much higher and persistent FVIII expression than the other 3 mice - is this related to anti-FVIII Ab development - it's not clear from Fig 6C. Also, why did any of these mice develop a FVIII immune response as prior studies with these transgenic animals have demonstrated strong immune tolerance to FVIII that is very difficult to break. How did that happen here?

We agree with the Reviewer that the FVIII R593C transgenic mouse line used in our work should remain immune tolerant following exposure to FVIII protein. We have further investigated the anti-FVIII immune response, by determining the FVIII neutralizing activity in the plasma of treated mice. The new data included in Fig. 6D of the revised manuscript show that only 1 out of 11 treated mice had a transient neutralizing antibody titer, spontaneously disappearing at later times. These data show that immune response is limited to the generation of low titer binding but not neutralizing anti-FVIII antibodies, without precluding FVIII persistence. We have included a description of the new data in the Results section of the revised manuscript. Please note that FVIII is a very immunogenic protein and even normal individuals may develop non-neutralizing anti-FVIII antibodies (Algiman PNAS 1992 PMID: 1570298; Meunier Blood Adv 2017 PMID: 29296830). Furthermore, please note that FVIII output in mice treated with GP64-LV varies between 5% and 15% of normal levels in this experiment, thus reflecting the typical variability in transduction and transgene expression of *in vivo* experiments. The levels of FVIII do not correlate with levels of anti-FVIII antibodies in this experiment.

g) Why are the levels of FVIII in the neonates so low and again there appears to be some evidence of a late break in tolerance in some animals.

We thank the Reviewer for this question. We have now repeated the experiment at higher LV doses and show sustained and robust expression of FVIII at around 100% of normal activity without signs of immune responses. Please note that animals treated as newborns (Fig 6H-P) are immune-competent hemophilia A mice and not immune-tolerant FVIII R593C transgenic mice used in the experiment shown in Fig 6A-F. We have now better specified this information in Figure 6 of the revised manuscript.

h) The LSECs studied in culture were obtained as immortalized cells (how were they immortalized?). The *ex vivo* phenotype of these cells is notoriously very fragile - how did the authors confirm a consistent LSEC phenotype in these studies?

We agree with the reviewer that the LSEC phenotype is difficult to maintain over time in culture. The LSEC used in our work are commercially available and we purchased them from Innoprot, which guarantees their quality (<https://innoprot.com/product/immortalized-human-hepatic-sinusoidal-endothelial-cells/>). The cells were immortalized with SV40 large T antigen and the company confirmed expression of von Willebrand Factor, CD31, and CD144 and a lack of expression of cytokeratin 18 and 19. In addition, to confirm the maintenance of these features, we performed a characterization reported in the new Expanded View Figure 6 of the revised manuscript. These cells express high CD31, CD144, and low CD34. The lack of expression of

CD45 and CD90 confirmed a stable endothelial phenotype. Importantly, the immunofluorescence confirmed the expression of FVIII, a typical marker of LSEC.

Overall, this reviewer needs to see more commentary/explanation of the inconsistencies reported in this manuscript. Currently the study conclusions re. enhanced LSEC tropism with GP64 are not adequately supported - this may be, in part, due to the presentation of the evidence.

Referee #2 (Remarks for Author):

In this manuscript, Milani and colleagues reported that using GP64-pseudotyped lentivirus (GP64-LV) transduced liver sinusoidal endothelial cells (LSECs) better than VSVG-LV and combining phagocytosis inhibitor CD47 could further improve the transduction efficiency of GP64-LV on LSECs. using GP64-LV to deliver FVIII transgene to LSECs could achieve stable therapeutic levels of FVIII and correct the bleeding phenotype of hemophilia A mice. There are concerns in this manuscript that need to be addressed.

Major concerns

- The title of this manuscript emphasizes the efficacy of GP64-LV transducing LSECs in hemophilia A gene therapy, but the data from hemophilia A model studies were very light and weak.

We thank the Reviewer for raising this point. To address the Reviewer's criticism, we treated additional newborn hemophilia A mice with VSV.G-LV or GP64-LV expressing FVIII at higher vector doses. We observed around 100% of normal FVIII activity in GP64-LV-treated mice which was 4-fold higher than in VSV.G-LV-treated mice. The new data included in Fig 6N-P of the revised manuscript strengthen our conclusion that GP64-pseudotyped LV allow achieving LSEC-derived long-term stable FVIII expression up to the fully normal range, following administration to both adult and newborn mice.

- In Figure 2, Figure 2B: were there statistically significant differences between those groups? The N was low. Were those three data points from three patients? Could they include all data points from 6 patients and increase N in 293T experiments?

We agree with the Reviewer that group size was low in this experiment. We now include more data from biological replicates and confirm more GFP spots in macrophages than 293T, independently of the presence of the envelope protein, indicating active capture of LV particles by macrophages.

Figure 2C shows the fresh serum opsonized phagocytosis efficiency of LV in macrophages. This result is contradictory to their previous report (Milani et al. 2017), which showed that complement components in fresh serum could inactivate LV.

We would like to point out that the opsonization assay we developed and described in the current manuscript is designed to detect phagocytosis. In contrast, the assay described in our previous report, also shown here in Figure 1, is designed to detect complement-mediated lysis. We report that the inactivation of LV in fresh serum is vector dose-dependent and saturated at high LV concentrations, such as at the dose used in Figure 2. Here we show that, in the presence of fresh serum, LV are opsonized, i.e., they become more susceptible to phagocytosis by primary human macrophages. This phenomenon is counteracted by the high surface content of CD47, thus

suggesting that CD47^{hi} LV remain protected from phagocytosis even when opsonized by anti-LV non-neutralizing antibody and complement components. To clarify this point, we have included a sentence in the Results section of the revised manuscript, which reads: “Note that, at the LV doses used in the phagocytosis assay, there is no detectable direct complement-mediated LV inactivation.” In addition, we have included more details in the Methods section and Figure legend.

- Figure 3D: The authors should provide untransduced liver tissue as a control for staining in parallel. Representative images represent how many biological repeats?

We thank the Reviewer for this suggestion and have now included an untransduced liver tissue slide as a negative control for the staining in Fig EV2 of the revised manuscript. The immune fluorescence images are representative of 5 mice *per* group. We have now added this information in the Figure legend of the revised manuscript.

- There was a discrepancy between the data shown in Figure 3A versus the text on page 7. There was no difference in VCN between the 2e10 and 4e10 GP64-LV groups, but it was described as an LV-dose-dependent increase.

We agree with the Reviewer and have now modified the text, according to the Reviewer’s request, in the revised manuscript.

- It may not be appropriate to use the VSVG-LV FIX data from their previous report, Milani et al. 2019, to compare the results from the current GP64-LV FIX study (figure 3E). It would be more rigorous if they conducted VSVG-LV and GP64-LV FIX studies in parallel.

We would like to point out that a pool of several different independent experiments is more representative than a single experiment even if performed in parallel, when comparing vectors with different features. However, to address the Reviewer’s criticism, we report a new dataset (different from our previous report) that includes data from 116 mice across multiple experiments performed with VSV.G-LV over the past several years in the revised manuscript.

- Fig EV3F: it is difficult to tell whether endothelial cells were transduced. It was unclear what was stained in those sections. The authors should provide some information for readers to understand what was stained in their immunohistochemistry images. Also, providing some zoom-in areas with higher magnification images may help.

We thank the Reviewer for this suggestion. We now clarify that immunohistochemistry is anti-GFP and we show higher magnification images in Fig EV3G of the revised manuscript.

- Figure 5: There were only 2 data points in many conditions, and some have huge variations between the two data points. The authors should increase the N number to {greater than or equal to}3 before drawing a conclusion.

We agree with the Reviewer. We have now performed additional experiments with human primary LSEC and have now included the data in Fig 5 E-H of the revised manuscript. The data overall confirm similar to slightly higher transduction of LSEC by GP64-LV compared to VSV.G-LV. In addition, we have revised the text to better reflect the available results in the revised manuscript.

- The authors stated that FVIII expression was detected in hemophilia A mice treated with GP64-LV-F8.coFVIII, but undetectable in animals injected with 5x10¹⁰TU/kg of VSVG-LV-F8co.FVIII. However, their previous study (Merlin et al. Mol Ther 2017) using VSVG-LV-F8

showed that all animals injected with 10^9 TU VSVG-LV-F8 had sustained FVIII expression. The author should commend the discrepancy in the results from their current vs. previous studies. If it was because of the quality of viruses made at different times, they should not use previous data from the VSVG-LV-FIX study to compare their current GP64-LV-FIX studies (Figure 3E).

We thank the Reviewer for raising this point. The LV construct used in the current study is different from the one used by Merlin et al. Specifically, here we use the endogenous F8 promoter, while Merlin et al. used a vascular E-cadherin promoter, we use a codon-optimized FVIII transgene sequence and include hematopoietic-cell-specific microRNA 142 target sequences. All these differences may explain the different outcomes observed in the two studies. We have now included this information in the Results section of the revised manuscript. Moreover, we would like to point out that the new data reported in Figure 6 of the revised manuscript now show FVIII expression by VSV.G-LV, although at lower levels than that obtained by GP64-LV.

- The authors stated that anti-FVIII antibodies did not neutralize FVIII activity (page 10, lines 6-7), but they only showed anti-FVIII total IgG (Figure 6C), which cannot distinguish the inhibitory vs. not-inhibitory antibodies. They did not provide inhibitor titers. Did the VSVG-LV group develop inhibitory antibodies?

We have now further investigated the anti-FVIII immune response, by determining the FVIII neutralizing activity in the plasma of treated mice. The new data included in Fig 6D of the revised manuscript show that only 1 out of 11 treated mice had a transient neutralizing antibody titer, spontaneously disappearing at later times. These data show that immune response is limited to the generation of low titer binding but not neutralizing anti-FVIII antibodies, without precluding FVIII persistence.

- In the neonatal model study, there was no significant difference in the survival rate in the hemostasis challenge between the GP64-LV group and the untreated or VSVG group. How long after the treatment the hemostasis challenge was performed? The authors may want to increase the number of mice or increase the dose of the virus to treat animals to strengthen their data to support their conclusion that GP64-LV works better than VSVG-LV.

We have followed the Reviewer's suggestion and have treated additional newborn hemophilia A mice with VSV.G-LV or GP64-LV expressing FVIII at higher vector doses. We observed around 100% of normal FVIII activity in GP64-LV-treated mice, which was 4-fold higher than in VSV.G-LV-treated mice. These data confirm the superiority of GP64-LV in the setting of LSEC-derived FVIII expression.

Minor concerns

- In Figure 1, Figure 1A: the legend for plot data was missing. Which one depicts VSVG-LV, and which one is GP64-LV?

We have now included the legend in Figure 1A.

Figure 1D, the Y-axis title does not make sense. It was unclear Fold of what?

We have now modified the Y-axis title to read "Fold infectious titer to 293T".

Figure E-I: it was unclear how long the incubation time was.

We now also include this information in the Figure legends, in addition to the Methods section.

- Fig EV1: gold particles per virion in the GP64-LV, CD47free GP64-LV, and control groups all were very low. Could the authors make a separate figure or two segments in Y-axis to zoom in on those low numbers so readers can get a better view of those data points?

We have followed the Reviewer's suggestion and split the Y-axis in Fig EV1 of the revised manuscript.

- Fig EV2D: the authors should provide panels from each fluorescent staining and then the merged image, as they did in Figure 3D, so that readers can see more detailed information about the transduction efficiency in each cell type.

Please note that we show the quantification of KC, LSEC, and hepatocytes among GFP-positive cells in Fig EV2 A-C. Please consider that showing 4 channels and 1 merge image for 6 experimental groups would result in 30 different images. We thus consider it more effective to show the merge image for each experimental group, as representative of the quantitative analysis. However, we provide larger images and at higher magnification, to facilitate visualization of each cell type.

- Fig EV4: it would be helpful to show the standard deviation in each group.

Please note that the standard error of the mean is so low that it is not visible in the graph, in this case.

- Page 22, "ReFACTO" should be changed to "Xyntha".

Please note that we have used "ReFACTO" or "Moroctocog alfa" as recombinant BDD FVIII (https://en.wikipedia.org/wiki/Moroctocog_alfa). We now include both names in the revised manuscript.

4th Apr 2024

Dear Dr. Cantore,

Thank you for the submission of your revised manuscript to EMBO Molecular Medicine. We have now heard back from the one referee who agreed to evaluate your manuscript. This referee also assessed author responses to concerns raised by the referee #1. I am pleased to inform you that we will be able to accept your manuscript pending the following final amendments:

1) Figures:

- Please upload individual, high-resolution files in TIFF, EPS or PDF format for each main and EV figures. Up to 5 EV figures should be uploaded as individual figure files, with their legends added to the manuscript text after the main figure legends. The remaining figures should be renamed "Appendix Figure S1" and S2 and stay compiled with their legends in a PDF labelled Appendix. The appendix file has a table of content with page numbers on the title page. For more information on figure presentation please check "Author Guidelines".

<https://www.embopress.org/page/journal/17574684/authorguide#datapresentationformat>

- Please add information in the legend of Figure EV2D that the images of 4e10 TU/kg are also presented in Figure 3D.

2) In the main manuscript file, please do the following:

- Please address all comments suggested by our data editors listed below:

o Figure legends:

1. Please note that information related to n is missing in the legends of figures 1d-e; 4d-e; EV 2a-c; EV 5b.

2. Please note that n=2 in figures EV 7a-d.

3. Please note that the error bars are not defined in the legends of figures 1d-e.

- Add up to 5 keywords.

- Rename "Conflict of interest" to "Disclosure and competing interests statement". We updated our journal's competing interests policy in January 2022 and request authors to consider both actual and perceived competing interests. Please review the policy <https://www.embopress.org/competing-interests> and update your competing interests if necessary. Please add the sentence "Luigi Naldini is a member of the journal's editorial board. This has no bearing on the editorial consideration of this article for publication."

- Author contributions: Please remove it from the manuscript and specify author contributions in our submission system. CRediT has replaced the traditional author contributions section because it offers a systematic machine-readable author contributions format that allows for more effective research assessment. You are encouraged to use the free text boxes beneath each contributing author's name to add specific details on the author's contribution. More information is available in our guide to authors:

<https://www.embopress.org/page/journal/17574684/authorguide#authorshippinguidelines>

- In data availability statement please remove provided text and add the sentence "This study includes no data deposited in external repositories."

3) Appendix: Please move the suppl. materials and methods to the "Methods" section of the main manuscript and reorganize the figures as suggested in point 1.

4) The Paper Explained: Please provide "The Paper Explained" and add it to the main manuscript text. Please check "Author Guidelines" for more information. <https://www.embopress.org/page/journal/17574684/authorguide#researcharticleguide>

5) Synopsis: Every published paper now includes a 'Synopsis' to further enhance discoverability. Synopses are displayed on the journal webpage and are freely accessible to all readers. They include separate synopsis image and synopsis text.

- Synopsis image: Please provide a striking image or visual abstract as a high-resolution jpeg file 550 px-wide x (250-400)-px high to illustrate your article.

- Synopsis text: Please provide a short standfirst (maximum of 300 characters, including space) as well as 2-5 one sentence bullet points that summarise the paper as a .doc file. Please write the bullet points to summarise the key NEW findings. They should be designed to be complementary to the abstract - i.e. not repeat the same text. We encourage inclusion of key acronyms and quantitative information (maximum of 30 words / bullet point). Please use the passive voice.

6) For more information: This space should be used to list relevant web links for further consultation by our readers. Could you identify some relevant ones and provide such information as well? Some examples are patient associations, relevant databases, OMIM/proteins/genes links, author's websites, etc...

7) As part of the EMBO Publications transparent editorial process initiative (see our Editorial at

<http://embomolmed.embopress.org/content/2/9/329>), EMBO Molecular Medicine will publish online a Review Process File (RPF) to accompany accepted manuscripts. This file will be published in conjunction with your paper and will include the anonymous referee reports, your point-by-point response and all pertinent correspondence relating to the manuscript. Let us know whether you agree with the publication of the RPF and as here, if you want to remove or not any figures from it prior to publication. Please note that the Authors checklist will be published at the end of the RPF.

8) Please provide a point-by-point letter INCLUDING my comments as well as the reviewer's reports and your detailed responses (as Word file).

I look forward to reading a new revised version of your manuscript as soon as possible.

Yours sincerely,

Zeljko Durdevic

*** Instructions to submit your revised manuscript ***

- 1) a .docx formatted version of the manuscript text (including Figure legends and tables)
- 2) Separate figure files*
- 3) supplemental information as Expanded View and/or Appendix. Please carefully check the authors guidelines for formatting Expanded view and Appendix figures and tables at <https://www.embopress.org/page/journal/17574684/authorguide#expandedview>
- 4) a letter INCLUDING the reviewer's reports and your detailed responses to their comments (as Word file).
- 5) The paper explained: EMBO Molecular Medicine articles are accompanied by a summary of the articles to emphasize the major findings in the paper and their medical implications for the non-specialist reader. Please provide a draft summary of your article highlighting
 - the medical issue you are addressing,
 - the results obtained and
 - their clinical impact.This may be edited to ensure that readers understand the significance and context of the research. Please refer to any of our published articles for an example.
- 6) For more information: There is space at the end of each article to list relevant web links for further consultation by our readers. Could you identify some relevant ones and provide such information as well? Some examples are patient associations, relevant databases, OMIM/proteins/genes links, author's websites, etc...
- 7) Author contributions: the contribution of every author must be detailed in a separate section.
- 8) EMBO Molecular Medicine now requires a complete author checklist (<https://www.embopress.org/page/journal/17574684/authorguide>) to be submitted with all revised manuscripts. Please use the checklist as guideline for the sort of information we need WITHIN the manuscript. The checklist should only be filled with page numbers where the information can be found. This is particularly important for animal reporting, antibody dilutions (missing) and exact values and n that should be indicated instead of a range.

9) Every published paper now includes a 'Synopsis' to further enhance discoverability. Synopses are displayed on the journal webpage and are freely accessible to all readers. They include a short stand first (maximum of 300 characters, including space) as well as 2-5 one sentence bullet points that summarise the paper. Please write the bullet points to summarise the key NEW findings. They should be designed to be complementary to the abstract - i.e. not repeat the same text. We encourage inclusion of key acronyms and quantitative information (maximum of 30 words / bullet point). Please use the passive voice. Please attach these in a separate file or send them by email, we will incorporate them accordingly.

You are also welcome to suggest a striking image or visual abstract to illustrate your article. If you do please provide a jpeg file 550 px-wide x 300-800px high.

10) A Conflict of Interest statement should be provided in the main text

11) Please note that we now mandate that all corresponding authors list an ORCID digital identifier. This takes <90 seconds to complete. We encourage all authors to supply an ORCID identifier, which will be linked to their name for unambiguous name identification.

Currently, our records indicate that the ORCID for your account is 0000-0002-9741-997X.

Link Not Available

Photos 400-800 DPI

*Additional important information regarding figures and illustrations can be found at

<https://bit.ly/EMBOPressFigurePreparationGuideline>. See also figure legend preparation guidelines:

<https://www.embopress.org/page/journal/17574684/authorguide#figureformat>

***** Reviewer's comments *****

Referee #2 (Comments on Novelty/Model System for Author):

The manuscript was significantly improved in this version.

Referee #2 (Remarks for Author):

Concerns have been addressed. The manuscript was significantly improved in this version.

The authors addressed the remaining editorial issues.

16th Apr 2024

Dear Dr. Cantore,

We are pleased to inform you that your manuscript is accepted for publication and is now being sent to our publisher to be included in the next available issue of EMBO Molecular Medicine.
